# Parameter-free Algorithms for the Stochastically Extended Adversarial Model

**Shuche Wang**[1]     **Adarsh Barik**[2]     **Peng Zhao**[3,4]     **Vincent Y. F. Tan**[1,5,6]

[1] Institute of Operations Research and Analytics, National University of Singapore
[2] Department of Computer Science and Engineering, Indian Institute of Technology Delhi
[3] National Key Laboratory for Novel Software Technology, Nanjing University
[4] School of Artificial Intelligence, Nanjing University
[5] Department of Mathematics, National University of Singapore
[6] Department of Electrical and Computer Engineering, National University of Singapore

shuche.wang@u.nus.edu, adarshbarik1@iitd.ac.in,
zhaop@lamda.nju.edu.cn, vtan@nus.edu.sg

## Abstract

We develop the first parameter-free algorithms for the Stochastically Extended Adversarial (SEA) model, a framework that bridges adversarial and stochastic online convex optimization. Existing approaches for the SEA model require prior knowledge of problem-specific parameters, such as the diameter of the domain $D$ and the Lipschitz constant of the loss functions $G$, which limits their practical applicability. Addressing this, we develop parameter-free methods by leveraging the Optimistic Online Newton Step (OONS) algorithm to eliminate the need for these parameters. We first establish a comparator-adaptive algorithm for the scenario with unknown domain diameter but known Lipschitz constant, achieving an expected regret bound of $\widetilde{O}\big(\|u\|_2^2 + \|u\|_2(\sqrt{\sigma_{1:T}^2} + \sqrt{\Sigma_{1:T}^2})\big)$, where $u$ is the comparator vector and $\sigma_{1:T}^2$ and $\Sigma_{1:T}^2$ represent the cumulative stochastic variance and cumulative adversarial variation, respectively. We then extend this to the more general setting where both $D$ and $G$ are unknown, attaining the comparator- and Lipschitz-adaptive algorithm. Notably, the regret bound exhibits the same dependence on $\sigma_{1:T}^2$ and $\Sigma_{1:T}^2$, demonstrating the efficacy of our proposed methods even when both parameters are unknown in the SEA model.

## 1 Introduction

We focus on online convex optimization (OCO) [1, 2, 3], a broad framework for sequential decision-making. In each round $t \in [T]$, a learner chooses a point $x_t$ from a convex set $\mathcal{X} \subseteq \mathbb{R}^d$. The environment then discloses a convex function $f_t : \mathcal{X} \to \mathbb{R}$, after which the learner incurs a loss $f_t(x_t)$ and updates their decision. The standard way to show the performance is via the *regret*, the total loss relative to a comparator $u \in \mathcal{X}$, defined as $\mathfrak{R}_T(u) = \sum_{t=1}^{T} f_t(x_t) - \sum_{t=1}^{T} f_t(u)$.

For convex problems, the regret can be bounded by $O(\sqrt{T})$ [4], which is known to be minimax optimal [5]. OCO encompasses two primary frameworks: adversarial OCO [4, 6], which aims to minimize regret against arbitrarily chosen loss functions, and stochastic OCO (SCO) [6, 7], which minimizes excess risk under i.i.d. losses. While both frameworks are well-studied, real-world scenarios typically fall between these theoretical extremes of purely adversarial or stochastic settings. The Stochastically Extended Adversarial (SEA) model proposed in [8] bridges the gap between traditional adversarial and stochastic frameworks in OCO. This hybrid approach serves as an intermediate formulation that captures aspects of both adversarial OCO and SCO settings.

39th Conference on Neural Information Processing Systems (NeurIPS 2025).

Table 1: Comparison of the regret bounds of existing results and our proposed algorithms.

| Algorithm | Free of $D$ | Free of $G$ | Bound on Expected Regret $\mathbb{E}[\mathfrak{R}_T(u)]$ |
|---|---|---|---|
| OFTLR, OMD (Sachs et al. [8], Chen et al. [12]) | ✗ | ✗ | $O\left(\sqrt{\sigma_{1:T}^2} + \sqrt{\Sigma_{1:T}^2}\right)$ |
| OONS (Theorem 3.2) | ✗ | ✗ | $\widetilde{O}\left(\sqrt{\sigma_{1:T}^2} + \sqrt{\Sigma_{1:T}^2}\right)$ |
| CA-OONS (Theorem 4.1) | ✓ | ✗ | $\widetilde{O}\left(\|u\|_2^2 + \|u\|_2(\sqrt{\sigma_{1:T}^2} + \sqrt{\Sigma_{1:T}^2})\right)$ |
| CLA-OONS (Theorem 4.5) | ✓ | ✓ | $\widetilde{O}\left(\|u\|_2^2(\sqrt{\sigma_{1:T}^2} + \sqrt{\Sigma_{1:T}^2}) + \|u\|_2^4 + \sqrt{\sigma_{1:T} + \mathfrak{G}_{1:T}}\right)$ |

Optimal performance in OCO, SCO, and SEA models typically relies on careful step-size tuning, which requires prior knowledge of problem parameters such as the diameter of the decision set and Lipschitz constants. However, these parameters are often unknown in practice, motivating the development of *parameter-free* algorithms that achieve comparable regret without requiring such oracle information. Specifically, parameter-free algorithms include *comparator-adaptive* algorithms (unknown diameter $D$) and *Lipschitz-adaptive* algorithms (unknown Lipschitz constant $G$). A related challenge arises when the decision set $\mathcal{X}$ is unbounded, allowing adversaries to induce arbitrarily large losses for linear functions. Traditional methods often circumvent this by assuming bounded domains, where $\sup_{x,y \in \mathcal{X}} \|x - y\|_2 \leq D$. Consequently, developing OCO algorithms that remain effective under both unknown parameters and unbounded domains is significantly more challenging than in classical settings [9, 10, 11].

To address these challenges, we propose "parameter-free" algorithms for the SEA model, accommodating potentially unbounded decision sets. Using the Optimistic Online Newton Step as our base algorithm, we systematically relax assumptions: first tackling the case of an unknown domain diameter $D$ (potentially infinite) with a known Lipschitz constant $G$, and then extending to the more complex scenario where both $D$ and $G$ are unknown. In the SEA model, at each time step $t$, the learner selects a distribution $\mathcal{D}_t$ over functions and incurs a loss $f_t(x_t)$, where $f_t$ is sampled from $\mathcal{D}_t$. The *expected gradient* is denoted as $\nabla F_t(x) = \mathbb{E}_{f_t \sim \mathcal{D}_t}[\nabla f_t(x)]$.

**Main Contributions.** Our main results and contributions are summarized as follows.

(1) We begin by introducing the Optimistic Online Newton Step (OONS) as our foundational algorithm. The OONS algorithm is inspired by [10]; however, we incorporate an adaptive step-size $\eta_t$ rather than a fixed step-size throughout the learning process. When the parameters $D$ and $G$ are known, we demonstrate that OONS achieves an expected regret bound of $\widetilde{O}(\sqrt{\sigma_{1:T}^2} + \sqrt{\Sigma_{1:T}^2})$, matching the state-of-the-art results in terms of dependence on the cumulative stochastic variance $\sigma_{1:T}^2$ and the cumulative adversarial variation $\Sigma_{1:T}^2$ [8, 12]. This establishes a solid foundation for our subsequent extensions to parameter-free algorithms.

(2) We introduce the first parameter-free (comparator-adaptive) algorithm for the SEA model that remains effective when the domain diameter $D$ is unknown, provided the Lipschitz constant $G$ is known. This is achieved through a meta-framework wherein each base learner operates within a distinct bounded domain, complemented by the Multi-scale Multiplicative-Weight with Correction (MsMwC) algorithm [10] for the meta-algorithm's weight updates. This construction yields an expected regret bound of $\widetilde{O}(\|u\|_2^2 + \|u\|_2(\sqrt{\sigma_{1:T}^2} + \sqrt{\Sigma_{1:T}^2}))$, where the bound scales with the $\ell_2$-norm of the comparator $u$ without requiring prior knowledge of the domain diameter $D$.

(3) We further consider a setting in which *both* the domain diameter $D$ and the Lipschitz constant $G$ are unknown. By devising appropriate update rules for the estimation of the domain diameter, we establish an expected regret bound of $\widetilde{O}(\|u\|_2^2(\sqrt{\sigma_{1:T}^2} + \sqrt{\Sigma_{1:T}^2}) + \|u\|_2^4 + \sqrt{\sigma_{1:T} + \mathfrak{G}_{1:T}})$ where $\sigma_{1:T}$ captures the deviation of the stochastic gradients (excluding squared norms), and $\mathfrak{G}_{1:T}$ denotes the sum of the maximum expected gradients over the sequence.

A summary of our results and the best existing results are included in Table 1. Due to space limitations, we hide the Lipschitz constant $G$ in the $\widetilde{O}(\cdot)$-notation in the regret bound of CLA-OONS algorithm.

## 1.1 Related Work

The SEA model [8] is motivated by foundational insights from the *gradual-variation online learning*. The study of gradual variation can be traced back to the works of [13] and [14], and it has gained significant traction in recent years [15, 16, 17, 18, 19]. Notably, the SEA model has emerged as a practical application of the gradual variation principle [16, 18, 19]. Furthermore, this model serves as a bridge between adversarial OCO and SCO. This intermediate framework is comprehensively understood in the context of expert prediction [20, 21] and the bandit setting [22, 23].

Parameter-free online learning has emerged as a fundamental advancement in machine learning, offering solutions to the critical challenge of parameter tuning in practice. In the baseline scenario, when both the diameter parameter $D$ and the gradient bound $G$ are known, algorithms leveraging Follow the Regularized Leader or Mirror Descent principles achieve the minimax optimal regret bound of $\mathfrak{R}_T(u) \leq O(GD\sqrt{T})$ [2]. The field has subsequently progressed to address more practical scenarios where complete parameter knowledge is unavailable. Notably, in the Lipschitz-adaptive setting, it is still possible to attain the same optimal regret bound, differing only by constant factors [24, 25]. Xie et al. [18] extended these principles to gradient-variation online learning.

In the comparator-adaptive setting, the online learning problem becomes substantially more challenging due to the unknown comparator's magnitude, which could cause the algorithm's predictions to significantly deviate from the optimal solution, leading to a large regret. For this challenging scenario, a key result has been established as $\mathfrak{R}_T(u) \leq \widetilde{O}(\|u\|_2 G\sqrt{T})$ [26, 24, 9, 27]. For scenarios where both parameters $D$ and $G$ are unknown, significant progress has been made recently. Cutkosky [25] developed an algorithm with $\mathfrak{R}_T(u) \leq \widetilde{O}(G\|u\|_2^3 + \|u\|_2 G\sqrt{T})$, while Mhammedi & Koolen [28] achieve $\mathfrak{R}_T(u) \leq \widetilde{O}(G\|u\|_2^3 + G\sqrt{\max_{t \leq T}(\sum_{s=1}^{t} \|g_s\|_2 / \max_{s \leq t} \|g_s\|_2)})$. An alternative approach by [29] presented the regret bound $\mathfrak{R}_T(u) \leq \widetilde{O}(\|u\|_2^2 G\sqrt{T})$. More recent advances including [11] achieve the regret $\mathfrak{R}_T(u) \leq \widetilde{O}(G\|u\|_2\sqrt{T} + L\|u\|_2^2\sqrt{T})$ under the condition that subgradients satisfy $\|g_t\|_2 \leq G + L\|x_t\|_2$. Cutkosky & Mhammedi [30] further improve it to $\widetilde{O}(G\|u\|_2\sqrt{T} + \|u\|_2^2 + G^2)$.

Besides parameter-free algorithms for OCO, [31] and [32] studied the parameter-free stochastic gradient descent (SGD) algorithms. Khaled et al. [33] introduced the concept of "tuning-free" algorithms, which achieve performance comparable to optimally-tuned SGD within polylogarithmic factors, requiring only approximate estimates of the relevant problem parameters.

Although this series of works on parameter-free algorithms in OCO provides valuable insights, these approaches cannot be directly applied to attain the optimal regret bounds for the SEA model without prior knowledge of parameters. This limitation stems from the fact that the desired bounds for the SEA model should be expressed in terms of the variance-like quantities $\sigma_{1:T}^2$ and $\Sigma_{1:T}^2$, rather than the time horizon $T$. While Sachs et al. [8] have attempted to address this issue by proposing an algorithm that adapts to an unknown strong convexity parameter, their step-size search range still depends on both $D$ and $G$, thereby restricting its fully parameter-free adaptivity.

## 2 Problem Setup and Preliminaries

In this section, we formulate the problem setup of the Stochastically Extended Adversarial (SEA) model, present the existing results, and discuss the key challenges.

### 2.1 Problem Setup of the SEA Model

In iteration $t \in [T]$, the learner selects a decision $x_t$ from a convex feasible domain $\mathcal{X} \subseteq \mathbb{R}^d$, and nature chooses a distribution $\mathcal{D}_t$ from a set of distributions over functions. Then, the learner suffers a loss $f_t(x_t)$, where $f_t$ is a random function sampled from the distribution $\mathcal{D}_t$. The distributions are allowed to vary over time, and by choosing them appropriately, the SEA model reduces to the adversarial OCO, SCO, or other intermediate settings. Additionally, for each $t \in [T]$, the *(conditional) expected function* is defined as $F_t(x) = \mathbb{E}_{f_t \sim \mathcal{D}_t}[f_t(x)]$ and the *expected gradient* is defined as $\nabla F_t(x) = \mathbb{E}_{f_t \sim \mathcal{D}_t}[\nabla f_t(x)]$. We define $\mathfrak{G}_t := \sup_{x \in \mathcal{X}} \|\nabla F_t(x)\|_2$ to be the largest norm of the expected gradient, and use the shorthand $\mathfrak{G}_{1:T}$ to denote the sum $\sum_{t=1}^{T} \mathfrak{G}_t$.

Due to the randomness in the online decision-making process, our goal in the SEA model is to bound the *expected regret* with respect to the randomness in the loss functions $f_t$ drawn from the distribution $\mathcal{D}_t$ against any fixed comparator $u \in \mathcal{X}$, defined as $\mathbb{E}[\mathfrak{R}_T(u)] \triangleq \mathbb{E}[\sum_{t=1}^{T} f_t(x_t) - \sum_{t=1}^{T} f_t(u)]$. To capture the characteristics of the SEA model, we introduce the following quantities. For each $t \in [T]$, define the *(conditional) variance of the gradients* and *cumulative stochastic variance* respectively as

$$\sigma_t^2 = \sup_{x \in \mathcal{X}} \mathbb{E}_{f_t \sim \mathcal{D}_t} \left[ \|\nabla f_t(x) - \nabla F_t(x)\|_2^2 \right], \quad \sigma_{1:T}^2 = \mathbb{E} \left[ \sum_{t=1}^{T} \sigma_t^2 \right], \tag{1}$$

which reflect the stochasticity of the online process. Additionally, we introduce the concepts of stochastic gradient deviation and cumulative gradient deviation to characterize the stochastic variation of gradients, without the squared norm. The stochastic gradient deviation is defined as $\sigma_t = \sup_{x \in \mathcal{X}} \mathbb{E}_{f_t \sim \mathcal{D}_t} \left[ \|\nabla f_t(x) - \nabla F_t(x)\|_2 \right]$, and the *cumulative gradient deviation* is defined as $\sigma_{1:T} = \mathbb{E} \left[ \sum_{t=1}^{T} \sigma_t \right]$. The *cumulative adversarial variation* is defined as

$$\Sigma_{1:T}^2 = \mathbb{E} \left[ \sum_{t=1}^{T} \sup_{x \in \mathcal{X}} \|\nabla F_t(x) - \nabla F_{t-1}(x)\|_2^2 \right],$$

where $\nabla F_0(x) = 0$, reflecting the adversarial difficulty. This work aims to provide expected regret bounds that depend on problem-dependent quantities such as $\sigma_{1:T}^2, \Sigma_{1:T}^2$, and $\mathfrak{G}_{1:T}$ instead of $T$.

Below, we present several assumptions. Note that our results do not rely on *all* of these assumptions; rather, specific assumptions are required for each result, which will be explicitly stated in the theorem.

**Assumption 2.1** (Boundedness of gradient norms). The gradient norms of all loss functions are bounded by $G$, i.e., $\max_{t \in [T]} \max_{x \in \mathcal{X}} \|\nabla f_t(x)\|_2 \leq G$.

**Assumption 2.2** (Boundedness of domain). The diameter of the convex set $\mathcal{X}$ (the feasible domain) is bounded by $D$ i.e., $\sup_{x,y \in \mathcal{X}} \|x - y\|_2 \leq D$.

**Assumption 2.3** (Smoothness). For all $t \in [T]$, the expected function $F_t$ is $L$-smooth over $\mathcal{X}$, i.e., $\|\nabla F_t(x) - \nabla F_t(y)\|_2 \leq L \|x - y\|_2$ for all $x, y \in \mathcal{X}$.

**Assumption 2.4** (Convexity). For all $t \in [T]$, the expected function $F_t$ is convex on $\mathcal{X}$.

**Notations.** Given a positive definite matrix $A$, the norm induced by $A$ is $\|x\|_A = \sqrt{x^\top A x}$. $\Delta_N$ denotes the $(N-1)$-dimensional simplex. Let $\psi : \mathcal{X} \to \mathbb{R}$ be a continuously differentiable and strictly convex function, the associated Bregman divergence is defined as $D_\psi(x, y) := \psi(x) - \psi(y) - \langle \nabla \psi(y), x - y \rangle$. The notation $O(\cdot)$ hides constants and $\widetilde{O}(\cdot)$ additionally hides polylog factors.

## 2.2 Existing Results for the SEA Model

**Bounded domain and gradient norm.** Sachs et al. [8] established a regret bound for the SEA model using both Optimistic Follow-The-Regularized-Leader (OFTRL) and Optimistic Mirror Descent (OMD), given by $\mathbb{E}[\mathfrak{R}_T(u)] = O(\sqrt{\sigma_{1:T}^2} + \sqrt{\Sigma_{1:T}^2})$, achieved by setting the step-size as $\eta_t = \frac{D^2}{\sum_{s=1}^{t-1} \min\{\frac{\eta_s}{2} \|g_s - m_s\|_2^2, D\|g_s - m_s\|_2\}}$, where $g_t = \nabla f_t(x_t)$ and $m_t = g_{t-1}$. Similarly, Chen et al. [12] derived the same bound by Optimistic Online Mirror Descent (OMD) with the step-size $\eta_t = \frac{D}{\sqrt{\delta + 4G^2 + \bar{V}_{t-1}}}$, where $\bar{V}_{t-1} = \sum_{s=1}^{t-1} \|g_s - m_s\|_2^2$ and $\delta > 0$.

In all of the above settings, the optimal step-size $\eta_t$ is dependent on the parameters $D$ (the diameter of decision set $\mathcal{X}$) and $G$ (Lipschitz constant), so there has been a natural motivation to develop algorithms that achieve similar regret bounds *without* knowing such parameters *a priori*. We term such algorithms as *"parameter-free"* algorithms for the SEA model.

**Parameter-free algorithm for the SEA model.** Theorem 5 in [27] demonstrates that the parameter-free mirror descent algorithm can be extended to enjoy a gradient-variation regret of $\mathfrak{R}_T(u) \leq \widetilde{O}(\|u\|_2 \sqrt{\sum_{t=1}^{T} \|\nabla f_t(x_t) - \nabla f_{t-1}(x_t)\|_2^2})$. In fact, this can directly yield an expected regret bound for the SEA model scaling with $\widetilde{\sigma}_{1:T}^2 := \mathbb{E} \left[ \sum_{t=1}^{T} \mathbb{E}_{f_t \sim \mathcal{D}_t} \left[ \sup_x \|\nabla f_t(x) - \nabla F_t(x)\|_2^2 \right] \right]$, i.e.,

$$\mathbb{E}[\mathfrak{R}_T(u)] \leq \widetilde{O} \left( \|u\|_2 \left( \sqrt{\widetilde{\sigma}_{1:T}^2} + \sqrt{\Sigma_{1:T}^2} \right) \right). \tag{2}$$

---

**Algorithm 1** Optimistic Online Newton Step (OONS)

---

**Input:** learning rate $\eta_t > 0$, $x_1' = 0$.

1: **for** $t = 1, \ldots, T$ **do**
2:    Receive optimistic prediction $m_t$ and range hint $z_t$.
3:    Update $x_t = \arg\min_{x \in \mathcal{X}}\{\langle x, m_t \rangle + D_{\psi_t}(x, x_t')\}$ where $\psi_t(x) = \frac{1}{2}\|x\|_{A_t}^2$ and $A_t = 4z_1^2 I + \sum_{s=1}^{t-1} \eta_s(\nabla_s - m_s)(\nabla_s - m_s)^\top + 4\eta_t z_t^2 I$.
4:    Receive $g_t = \nabla f_t(x_t)$ and construct $\nabla_t = g_t + 32\eta_t \langle x_t, g_t - m_t \rangle (g_t - m_t)$.
5:    Update $x_{t+1}' = \arg\min_{x \in \mathcal{X}}\{\langle x, \nabla_t \rangle + D_{\psi_t}(x, x_t')\}$.
6: **end for**

---

Akin to $\sigma_{1:T}^2$, $\widetilde{\sigma}_{1:T}^2$ defined in [12] also captures the stochastic nature of the SEA model. Furthermore, in the worst case, the bound in (2) reduces to $\widetilde{O}(\|u\|_2 \sqrt{T})$, matching the best available problem-independent bound. The outer expectation in the definition of $\widetilde{\sigma}_{1:T}^2$ accounts for the randomness in the choice of the distribution $\mathcal{D}_t$ at each step. Refer to Appendix B.1 for a self-contained proof of (2).

**Key Challenge.** However, we emphasize that our goal is to obtain regret bounds scaling with $\sigma_{1:T}^2$, as defined in (1). As pointed out in previous work on the SEA model [12, Remark 9], $\sigma_{1:T}^2$ is more favorable than $\widetilde{\sigma}_{1:T}^2$. First, from a mathematical perspective, the latter is generally larger due to the convexity of the supremum operator. The difference between $\sigma_{1:T}^2$ and $\widetilde{\sigma}_{1:T}^2$ can, in fact, be arbitrarily large. The detailed comparison is provided in Appendix A. Second, from an algorithmic perspective, an algorithm with a regret bound involving $\widetilde{\sigma}_{1:T}^2$ typically involves an *implicit* update, which operates on the original function and is significantly more costly than standard first-order methods (see Remark 10 in [12]). Third, achieving regret bounds scaling with $\sigma_{1:T}^2$ typically requires leveraging the *Regret Bounded by Variation in Utilities (RVU)* property [34], which captures the regret to be bounded not only by the gradient variations but also an additional negative stability term. Formally, an algorithm satisfies the RVU property if its regret upper bound is in the form of $\sum_{t=1}^T \langle x^* - x_t, u_t \rangle \leq \alpha + \beta \sum_{t=1}^T \|u_t - u_{t-1}\|^2 - \gamma \sum_{t=1}^T \|x_t - x_{t-1}\|^2$, for some constants $\alpha, \beta, \gamma > 0$. This structure enables finer control over the regret by explicitly analyzing trajectory stability, establishing profound connections to game theory [34, 35] and accelerations in smooth optimization [36]. Consequently, the key challenge lies in how to achieve this preferred $\sigma_{1:T}^2$-scaling *without* knowledge of $D$ and $G$ for unbounded domains.

## 3 Optimistic Online Newton Step (ONS) for the SEA Model

In this section, different from the Optimistic follow-the-regularized-leader (OFTRL) [8] and Optimistic mirror descent (OMD) [12], we first introduce the Optimistic Online Newton Step (OONS) algorithm as the base algorithm for the "parameter-free" algorithms to be introduced later. This algorithm is summarized in Algorithm 1.

The ONS algorithm [6] is an iterative algorithm that adaptively updates a second-order (Hessian-based) model of the loss, allowing more efficient gradient-based updates and improved regret bounds. OONS also maintains two sequences $\{x_t\}_{t=1}^T$ and $\{x_t'\}_{t=1}^T$ like OMD and OFTRL, which is achieved by introducing the optimistic prediction $m_t$. Chen et al. [10] also considered combining their Multi-scale Multiplicative-weight with Correction (MsMwC) algorithm with this variant of the ONS algorithm. However, the step-size $\eta$ is fixed in their algorithm and the MsMwC algorithm is applied to learn the optimal $\eta_\star$. Different from it, in OONS, we consider adaptive step-sizes $\eta_t$.

**Theorem 3.1.** *Suppose that $\|g_t - m_t\|_2 \leq z_t$, $z_t$ is non-decreasing in $t$, $64\eta_t D z_T \leq 1$ for all $t \in [T]$, and $\eta_t$ is non-increasing in $t$. Then,* OONS *guarantees that*

$$\mathfrak{R}_T(u) \leq O\left( \frac{r \ln(T \eta_1 z_T / z_1)}{\eta_T} + z_1^2 \|u\|_2^2 + D(z_T - z_1) + \sum_{t=1}^T \eta_t \langle u, g_t - m_t \rangle^2 - z_1^2 \sum_{t=2}^T \|x_t - x_{t-1}\|_2^2 \right) \tag{3}$$

*where $r$ is the rank of $\sum_{t=1}^T (g_t - m_t)(g_t - m_t)^\top$.*

Next, we verify that OONS also works for the case with known parameters $D, G$, and we can also obtain a similar regret bound as [8] and [12]. The regret bound of OONS for the SEA model with

---

**Algorithm 2** Comparator-adaptive algorithm for the SEA model (CA-OONS)

---

**Input:** Lipschitz constant $G$.

1: **for** $t = 1, \ldots, T$ **do**
2:    Create $N = \lceil \log T \rceil$ base-learners. Each base-learner $j \in [N]$ runs OONS with stepsize $\eta_t^j$.
3:    Each base-learner $j$ provides $x_t^j$.
4:    Run Algorithm 3 to obtain $w_t \in \Delta_N$.
5:    The final decision is $x_t = \sum_{j=1}^N w_{t,j} x_t^j$.
6: **end for**

---

known parameters $D$ and $G$ is presented below. We specify the adaptive step-size for all $t \in [T]$ as

$$\eta_t = \min \left\{ \frac{1}{64 D z_T}, \frac{1}{D \sqrt{\sum_{s=1}^{t-1} \| g_s - m_s \|_2^2}} \right\}. \tag{4}$$

Since $G$ is known, we have $\| g_t - m_t \|_2 \le z_t = 2G, \forall t \in [T]$ and $\eta_t$ is defined in terms of $z_T$ here.

**Theorem 3.2.** *Under Assumptions 2.1, 2.2, 2.3, and 2.4,* OONS *with step-size $\eta_t$ given in* (4), $m_t = \nabla f_{t-1}(x_{t-1})$ *and $z_t = 2G$ for all $t \in [T]$ ensures $\mathbb{E}[\mathfrak{R}_T(u)] = \widetilde{O}(\sqrt{\sigma_{1:T}^2} + \sqrt{\Sigma_{1:T}^2})$.*

*Remark* 3.3. Theorem 3.2 achieves the same (up to logarithmic terms) dependence on $\sigma_{1:T}^2$ and $\Sigma_{1:T}^2$ as in [8] and [12]. The primary reason to use OONS as the base algorithm instead of OMD [12] is that the final regret bound for OMD typically depends on $D \sqrt{\sum_{t=1}^T \| g_t - m_t \|_2^2}$. In scenarios when $D$ is unknown or potentially infinite, like in Section 4.2, this might lead to $O(T)$ regret bounds. By contrast, OONS leverages adaptive second-order information, which helps remove (or substantially reduce) explicit dependence on $D$. In (3), the only term relevant to $D$ is $D(z_T - z_1)$, which solely depends on the starting and ending points, $z_1$ and $z_T$.

## 4 Parameter-free Algorithms for the SEA Model

In this section, we develop parameter-free algorithms for the SEA model, building on OONS which we use as the base algorithm. Moreover, we allow the decision set $\mathcal{X}$ to be potentially unbounded throughout this section, i.e. $D = \infty$ in Assumption 2.2. In Section 4.1, we present a *comparator-adaptive* algorithm for the SEA model for unknown $D$ but known $G$, and then, we develop the *comparator- and Lipschitz-adaptive* algorithm where both $D$ and $G$ are unknown in Section 4.2.

### 4.1 Comparator-adaptive algorithm

We now propose the Comparator-Adaptive Optimistic Online Newton Step (CA-OONS) algorithm for the unknown $D$ (potentially infinite) but known $G$ case by using a meta-base algorithm framework. Recent works in [10] and [11] addressed the challenges associated with unbounded domains by developing a base-learner framework. Building on this philosophy, we propose CA-OONS(Algorithm 2) where we adopt the MsMwC–Master algorithm [10] as the meta algorithm.

The algorithm uses $N$ base-learners. For *any* base-learner $i \in [N]$, the regret can be decomposed as

$$\mathfrak{R}_T(u) = \sum_{t=1}^T f_t(x_t) - \sum_{t=1}^T f_t(u) = \underbrace{\sum_{t=1}^T f_t(x_t) - \sum_{t=1}^T f_t(x_t^i)}_{\text{Meta Regret}} + \underbrace{\sum_{t=1}^T f_t(x_t^i) - \sum_{t=1}^T f_t(u)}_{\text{Base Regret}} .$$

Let $\mathcal{A}_i$ denote the base algorithm for the $i$-th base-learner. We denote $\mathfrak{R}_T^{\mathcal{A}_i}(u) = \sum_{t=1}^T f_t(x_t^i) - \sum_{t=1}^T f_t(u)$ as the base regret by taking $\mathcal{A}_i$ as the base algorithm. Moreover, the final decision $x_t$ is a weighted-average of all the base-learners' decisions: $x_t = \sum_{j=1}^N w_{t,j} x_t^j$ with $w_t \in \Delta_N$. As such,

$$\mathfrak{R}_T(u) \le \mathfrak{R}_T^{\mathcal{A}_i}(u) + \sum_{t=1}^T \langle \ell_t, w_t - w_\star^i \rangle, \tag{5}$$

**Algorithm 3** Meta Algorithm

---

**Input:** Additional expert set $\mathcal{S}$ defined in (7).
**Initialization:** $p_1' \in \Delta_{\mathcal{S}}$ such that $p_{1,k}' \propto \beta_k^2$ for all $k \in \mathcal{S}$.

1: **for** $t = 1, \ldots, T$ **do**
2:     Construct $h_t \in \mathbb{R}^N$ with $h_t^j = \langle \nabla f_{t-1}(x_{t-1}^j), x_t^j \rangle$.
3:     Each expert $k \in \mathcal{S}$ runs MsMwC with step-size $\beta_{t,j}^k$ and plays $w_t^k \in \Delta_N$.
4:     Receive $w_t^k$ for each $k \in \mathcal{S}$ and compute $H_t^k = \langle w_t^k, h_t \rangle$.
5:     Compute $p_t = \arg\min_{p \in \Delta_{\mathcal{S}}} \langle p, H_t \rangle + D_\phi(p, p_t')$.
6:     Play $w_t = \sum_{k \in \mathcal{S}} p_{t,k} w_t^k \in \Delta_N$.
7:     Receive $\ell_t \in \mathbb{R}^N$. Define $L_t^k = \langle w_t^k, \ell_t \rangle$ and $b_t^k = 32\beta_k(L_t^k - H_t^k)$.
8:     Compute $p_{t+1}' = \arg\min_{p \in \Delta_{\mathcal{S}}} \langle p, L_t + b_t \rangle + D_\phi(p, p_t')$.
9: **end for**

---

Table 2: Three-layer hierarchy of CA-OONS

| **Layer** | **Algorithm** | **Loss** | **Optimism** | **Decision** | **Output** |
|---|---|---|---|---|---|
| **Top Meta** | MsMwC | $(L_t^k)_{k \in \mathcal{S}}$ | $(H_t^k)_{k \in \mathcal{S}}$ | $p_t \in \Delta_{\mathcal{S}}$ | $w_t = \sum_k p_{t,k} w_t^k$ |
| **Middle Meta** | MsMwC | $\ell_t \in \mathbb{R}^N$ | $h_t \in \mathbb{R}^N$ | $(w_t^k)_{k \in \mathcal{S}} \in \Delta_N$ | $w_t^k$ |
| **Base** | OONS | $\nabla f_t(x_t^j)$ | $\nabla f_{t-1}(x_{t-1}^j)$ | $(x_t^j)_{j \in N} \in \mathcal{X}_j$ | $x_t^j$ |

where $\ell_t \in \mathbb{R}^N$ with $\ell_t^j = \langle \nabla f_t(x_t^j), x_t^j \rangle$ and $w_\star^i$ is a vector in $\Delta_N$ whose $j$-th component is $(w_\star^i)_j = 1$ if $j = i$ and 0 otherwise. Refer to Appendix D.2 for the proof of (5).

We first consider the base algorithm. Specifically, for each base-learner $j \in [N]$, we impose a constraint that it operates within $\mathcal{X}_j = \{x : \|x\|_2 \leq D_j \text{ and } x \in \mathcal{X}\}$, where $D_j = 2^j$. Then, we define $g_t^j = \nabla f_t(x_t^j)$ and $m_t^j = \nabla f_{t-1}(x_{t-1}^j)$. Each base-learner $j \in [N]$ runs OONS with step-size

$$\eta_t^j = \min \left\{ \frac{1}{64 D_j z_T}, \frac{1}{D_j \sqrt{\sum_{s=1}^{t-1} \|g_s^j - m_s^j\|_2^2}} \right\}, \tag{6}$$

which depends on $D_j$ instead of $D$ in OONS. Hence, each base-learner $j$ can update $x_t^j$ via OONS with step-size $\eta_t^j$. Since the final decision is $x_t = \sum_{j=1}^N w_{t,j} x_t^j$, we need to adopt a meta-algorithm to learn the weight parameter $w_t \in \Delta_N$.

As mentioned above, we introduce a constraint that each base-learner $j \in [N]$ operates within a $D_j$-bounded domain. We can consider this as a "multi-scale" base-learner problem [37, 9, 10] where each base-learner $j$ has a different loss range such that $|\ell_t^j| \leq G D_j$ since $\ell_t^j = \langle \nabla f_t(x_t^j), x_t^j \rangle$. We choose the Multi-scale Multiplicative-weight with Correction (MsMwC)–Master algorithm (Algorithm 2 in [10]) as the meta-algorithm to learn $w_t$, which is implemented based on the MsMwC [10]. Details of MsMwC are presented in Appendix D.1.Specifically, we define a new expert set

$$\mathcal{S} = \{k \in \mathbb{Z} : \exists j \in [N], G D_j \leq 2^{k-2} \leq G D_j \sqrt{T}\}. \tag{7}$$

For all $k \in \mathcal{S}$, the step-size of the MsMwC–Master algorithm is set to $\beta_k = \frac{1}{32 \cdot 2^k}$. Each expert $k \in \mathcal{S}$ runs the MsMwC algorithm with $w_1'$ being uniform over $\mathcal{Z}(k)$, where $\mathcal{Z}_k = \{j \in [N] : G D_j \leq 2^{k-2}\}$. Moreover, each base MsMwC algorithm only works in the subset $\mathcal{Z}(k)$, i.e., $w_t \in \Delta_N$ with $w_{t,j} = 0$ for all $j \notin \mathcal{Z}(k)$. We can view CA-OONS (Algorithm 2) as a three-layer structure, where the meta-algorithm (Algorithm 3) itself consists of two layers, which we refer to as *meta top* and *meta middle*. Also, the base layer is OONS algorithm. For clarity, we summarize the notations of the three-layer hierarchy for CA-OONS in Table 2.

In the following, we provide the expected regret guarantee for CA-OONS.

**Theorem 4.1.** *Let $D$ be unknown (potentially infinite). Under Assumptions 2.1, 2.3 and 2.4,* CA-OONS *provides the following regret*

$$\mathbb{E}[\mathfrak{R}_T(u)] = \widetilde{O}\Big(\|u\|_2^2 + \|u\|_2(\sqrt{\sigma_{1:T}^2} + \sqrt{\Sigma_{1:T}^2})\Big). \tag{8}$$

This regret guarantee is referred to as "comparator-adaptive" because it depends directly on the norm of the comparator, $\|u\|_2$, rather than explicitly relying on the diameter of the decision set, $D$. Notably, when considering the constrained decision set with a diameter $D$, our regret bound immediately recovers the result $\mathbb{E}[\mathfrak{R}_T(u)] = \widetilde{O}(D^2 + D(\sqrt{\sigma_{1:T}^2} + \sqrt{\Sigma_{1:T}^2}))$ established in [8, 12].

One limitation of our regret bound (8) is that when particularizing to adversarial OCO, it achieves only an $\widetilde{O}(\|u\|_2^2 + \|u\|_2\sqrt{T})$ worst-case regret bound, which falls short of the best-known $\widetilde{O}(\|u\|_2\sqrt{T})$ regret bound [24, 27, 28]. However, in the following two remarks, we will justify the $\|u\|_2$-dependence in the gradient-variation regret and emphasize the fundamental challenge of achieving adaptivity from the gradient-variation bound (for smooth functions) to the worst-case bound (for the non-smooth case) when the decision set of online learning is *unconstrained*.

*Remark* 4.2 (Dependency on $\|u\|_2^2$). Recent studies have established the connection between gradient-variation online learning and accelerated offline optimization through advanced online-to-batch conversions [36, 38]. Specifically, let $d_0 = \|x_0 - x_*\|_2$ denote the distance of an initial point $x_0$ to the optimum $x_*$. For an $L$-smooth function, gradient-variation online algorithms using first-order information correspond to an accelerated convergence rate of $O(Ld_0^2/T^2)$ via the *stabilized* online-to-batch conversion [39]. For a $G$-Lipschitz function, the problem-independent regret bounds translate to an $O(Gd_0/\sqrt{T})$ rate through the standard conversion [1]. In this context, we hypothesize that the $\|u\|_2^2$ term may be unavoidable in gradient-variation regret for unconstrained online learning, paralleling how the $d_0^2$ term also appears in the accelerated rate of unconstrained offline optimization.

*Remark* 4.3 (Adaptivity between gradient-variation bound and worst-case bound). We argue that achieving *adaptivity* between the gradient-variation bound and the problem-independent worst-case bound in *unconstrained* online learning may be as challenging as achieving *universality* in offline optimization over unconstrained domains, where the method must adapt to both smooth and Lipschitz functions. To the best of our knowledge, the best-known universal method for offline unconstrained optimization is by [40], which combines UNIXGRAD [39] with the DoG step size [31]. Nonetheless, this method is complex and still relies on a predefined range of parameters, highlighting both the difficulty of the problem and the fact that it remains only partially solved. Consequently, designing a *single* unconstrained online learning algorithm that adaptively bridges the gradient-variation regret bound for smooth functions and the worst-case bound for Lipschitz functions is non-trivial, which could provide new insights into universal offline optimization methods. We leave this for future work.

*Remark* 4.4 (On dependence on the time horizon). The use of $T$ in CA-OONS (via $N = \lceil \log T \rceil$ experts) is only for convenience and not fundamental. An *anytime* variant is obtained by the standard doubling trick: restart the algorithm at epochs of lengths $1, 2, 4, \ldots$, and in epoch $k$ set $N_k = \lceil \log 2^{k-1} \rceil$. This introduces at most an additional logarithmic factor already hidden in $\widetilde{O}(\cdot)$ [41, Section 4.3]. A restart-free alternative is a sleeping (awakening) expert grid of learning rates as in the multi–rate construction of [42], which activates only those experts whose scale becomes relevant.

## 4.2 Comparator- and Lipschitz-adaptive Algorithm

The algorithm in the previous subsection requires *prior* knowledge of the Lipschitz constant $G$. Due to practical limitations such knowledge may not be available in real applications. A comparator- and Lipschitz-adaptive algorithm would instead *adapt* to an unknown Lipschitz constant $G$.

A simple approach to handling the unknown gradient norms, proposed by [25], relies on a gradient-clipping reduction. The key idea is to design an algorithm $\mathcal{A}$ that achieves appropriate regret when given prescient "hints" $h_t \geq \|g_t\|_2$ at the start of round $t$. Since such hints are impractical (as $g_t$ is not observed beforehand), we instead approximate them using a clipped gradient, inspired by [25]. We start with an initial guess $B_0$ on the range of $\max_t \|g_t - m_t\|_2$, where $g_t = \nabla f_t(x_t)$. We define $B_t = \max_{0 \leq s \leq t} \|g_s - m_s\|_2$ as the predicted error range up to iteration $t$. The truncated gradient is then defined as $\widetilde{g}_t = m_t + \frac{B_{t-1}}{B_t}(g_t - m_t)$. The truncated gradient satisfies $\|\widetilde{g}_t - m_t\|_2 \leq B_{t-1}$, allowing the learner to assume that the range of predicted error in iteration $t$ is known at the start.

Next, we initialize the decision set diameter guess as $D_1 = 1$. For each iteration $t \in [T]$, we first play $x_t$ and receive $g_t = \nabla f_t(x_t)$. To update $D_t$, we consider the condition $D_t <$

---

**Algorithm 4** Comparator and Lipschitz-Adaptive (or CLA-OONS) for the SEA model

---

**Input:** Initial scale $B_0$.

**Initialize:** $D_1 = 1$.

1: **for** $t = 1, \dots, T$ **do**
2:     Run OONS in $D_t$-bounded domain and obtain $x_t$. Play $x_t$ and receive $g_t = \nabla f_t(x_t)$.
3:     Construct $\widetilde{g}_t = m_t + \frac{B_{t-1}}{B_t}(g_t - m_t)$, where $B_t = \max_{0 \le s \le t} \|g_s - m_s\|_2$.
4:     **if** $D_t < \sqrt{\sum_{s=1}^{t} \frac{\|g_s\|_2}{\max\{1, \max_{k \le s} \|g_k\|_2\}}}$ **then**
5:         Update $D_{t+1} = 2\sqrt{\sum_{s=1}^{t} \frac{\|g_s\|_2}{\max\{1, \max_{k \le s} \|g_k\|_2\}}}$ and reset $A_{t+1}$ as (9) and $x'_{t+1} = 0$.
6:     **end if**
7:     Feed $\widetilde{g}_t$ to OONS running in the $D_{t+1}$-bounded domain and get $x_{t+1}$, where $z_{t+1} = B_t$.
8: **end for**

---

$\sqrt{\sum_{s=1}^{t} \frac{\|g_s\|_2}{\max\{1, \max_{k \le s} \|g_k\|_2\}}}$. If this condition holds, we update $D_{t+1}$ using the doubling trick. This ensures that we need to update $D_t$ a maximum of $M = O(\log T)$ times. We divide the total $T$ iterations into disjoint subsets of $M$ iterations. If the "doubling" occurs at the $t$-th iteration, we update $t_a \leftarrow t$ and reset $x'_{t+1} = 0$ and the matrix $A_{t+1}$ in OONS as follows

$$A_{t+1} = 4z_{t_a+1}^2 I + \sum_{s=t_a+1}^{t} \eta_s (\nabla_s - m_s)(\nabla_s - m_s)^\top + 4\eta_{t+1} z_{t+1}^2 I. \tag{9}$$

Then, we feed $\widetilde{g}_t$ to OONS running in the $D_{t+1}$-bounded domain $\mathcal{X}_{t+1} = \{x : \|x\|_2 \le D_{t+1} \wedge x \in \mathcal{X}\}$ and obtain $x_{t+1}$. We summarize the ideas in Algorithm 4 and term it as Comparator and Lipschitz-Adaptive Optimistic Online Newton Step (or CLA-OONS) algorithm.

**Theorem 4.5.** *Let both $D$ (potentially infinite) and $G$ be unknown. Under Assumptions 2.1 (but $G$ is unknown), 2.3 and 2.4, the proposed* FPF-OONS *algorithm satisfies*

$$\mathbb{E}[\mathfrak{R}_T(u)] \le \widetilde{O}\Big(\|u\|_2^2(\sqrt{\sigma_{1:T}^2} + \sqrt{\Sigma_{1:T}^2}) + G^2\|u\|_2^2 + \|u\|_2^4 + G\|u\|_2^3 + G^2\sqrt{\sigma_{1:T} + \mathfrak{G}_{1:T}}\Big),$$

*where $\sigma_{1:T}$ captures the stochastic gradient deviation (without the squared norm) and $\mathfrak{G}_{1:T}$ denotes the sum of maximum expected gradients.*

*Remark* 4.6 (Discussion and challenges). In Theorem 4.5, the regret includes $\|u\|_2^2(\sqrt{\sigma_{1:T}^2} + \sqrt{\Sigma_{1:T}^2})$. Ideally, we aim to achieve a dependence of $\widetilde{O}(\|u\|_2)$, consistent with [25] and [43]. However, achieving this within the SEA framework presents significant challenges. As mentioned in Section 2, obtaining regret bounds that scale with $\sigma_{1:T}^2$ in the SEA framework is difficult. These challenges are compounded in the comparator and Lipschitz-adaptive setting. Below, we outline some of the main technical challenges associated with achieving the desired bound of $\widetilde{O}(\|u\|_2(\sqrt{\sigma_{1:T}^2} + \sqrt{\Sigma_{1:T}^2}))$.

As stated in Section 2, the methods such as those proposed by [25, 27, 11] cannot be applied to obtain the expected regret bound in terms of $\sigma_{1:T}^2$. The work [44] also looks promising; however, it remains unclear to us whether the approach proposed in the paper can be directly extended to the SEA framework. Their results, presented in Theorems 3 and 5 of [44], are not Lipschitz-adaptive. Specifically, they operate under the assumptions that $\|g_t\|_2 \le 1$ and $\|m_t\|_2 \le 1$.

One could also use a large number of base-learners to achieve a regret of $\widetilde{O}(\|u\|_2 \sqrt{r \sum_t \|g_t - m_t\|_2^2} + \|u\|_2^3)$, similar to [10]. However, this approach presents a subtle yet significant challenge. Following a similar analysis [10], we get the following decomposition: $\sum_t \langle g_t, x_t - u \rangle = \sum_t \langle g_t, x_t - x_t^{k_*} \rangle + \sum_t \langle g_t, x_t^{k_*} - u \rangle$. By leveraging Theorem 23 in [10], we can write $\sum_t \langle g_t, x_t^{k_*} - u \rangle$ as $\sum_t \langle g_t, x_t^{k_*} - u \rangle \le \widetilde{O}(\|u\| \sqrt{r \sum_t \|g_t - m_t\|_2^2} - \sum_t \|x_t^{k_*} - x_{t-1}^{k_*}\|_2^2)$. Observe that expressing the first term, $\sqrt{r \sum_t \|g_t - m_t\|_2^2}$, in terms of $\sigma_{1:T}$ and $\Sigma_{1:T}$ introduces additional terms involving $\sum_t \|x_t - x_{t-1}\|_2^2$ (See Lemma 2 in [12]). The only way to address this term is through the negative term $-\sum_t \|x_t^{k_*} - x_{t-1}^{k_*}\|_2^2$, which becomes tricky. This challenge is reminiscent of the problem encountered by [17]. Their solution, as outlined in Equation (17) of [17], relies on the bounded domain assumption, which is not applicable in our setting. Consequently, this limitation prevents us from improving the $\|u\|_2^2$ dependence in the leading term by using additional base-learners.

The bound in Theorem 4.5 also includes an additive term involving $\sqrt{\mathfrak{G}_{1:T}}$, which reflects the sum of the maximum expected gradients' norms over $T$ rounds, and arises because the domain is potentially unbounded. Note that this term does not have a $\|u\|$ dependence. Hence, the comparator having a large norm in an unbounded setting (potentially dependent on $T$) does not affect its growth. In the worst case, $\sqrt{\mathfrak{G}_{1:T}} = O(\sqrt{T})$, which underscores that this additive term does not have a significant adverse effect on the regret as $\sqrt{\sigma_{1:T}^2}$ and $\sqrt{\Sigma_{1:T}^2}$ also scale as $\sqrt{T}$ [8].

## 5   Conclusions and Future Work

This paper presents novel parameter-free algorithms for the SEA model, addressing critical challenges in online optimization where traditional approaches require prior knowledge of parameters such as the diameter of the domain $D$ and the Lipschitz constant of the loss functions $G$. Our proposed algorithms: CA-OONS and CLA-OONS are designed to operate effectively even when $D$ and $G$ are unknown, demonstrating their adaptability and practicality.

There are several avenues for future research. First, we would like to improve the regret's dependence on $\|u\|_2$ when both $D$ and $G$ are unknown. Another promising direction is to reduce the number of gradient queries in CA-OONS from $O(\log T)$ to $O(1)$, thus enhancing its efficiency. An intriguing question in the comparator-adaptive setting is whether it is possible to design a single, simple online algorithm that simultaneously achieves two types of bounds: $\widetilde{O}(\|u\|_2^2 + \|u\|_2(\sqrt{\sigma_{1:T}^2} + \sqrt{\Sigma_{1:T}^2}))$ and $\widetilde{O}(\|u\|_2 G \sqrt{T})$. As discussed in Remark 4.3, it remains an open challenge to construct an adaptive parameter-free online algorithm that can interpolate between these bounds. An additional open direction is to move beyond *expected* regret and derive *high-probability* (or variance-sensitive) regret guarantees for the SEA model in the parameter-free setting. Current SEA analyses, including [8, 12], bound only $\mathbb{E}[\mathfrak{R}_T(u)]$; developing concentration results that retain the fine $\sigma_{1:T}^2$ and $\Sigma_{1:T}^2$ dependence without incurring suboptimal logarithmic inflation appears non-trivial and is left for future work.

## Acknowledgments and Disclosure of Funding

This research is funded by the Singapore Ministry of Education Academic Research Fund Tier 2 under grant number A-8000423-00-00 and three Singapore Ministry of Education Academic Research Funds Tier 1 under grant numbers A-8000189-01-00, A-8000980-00-00 and A-8002934-00-00.

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

# A   Separation between $\sigma_{1:T}^2$ and $\widetilde{\sigma}_{1:T}^2$

Recall (Section 2) the two stochastic variance measures:

$$\sigma_t^2 = \sup_{x \in \mathcal{X}} \mathbb{E}\big[\|\nabla f_t(x) - \nabla F_t(x)\|_2^2\big], \qquad \widetilde{\sigma}_t^2 = \mathbb{E}\Big[\sup_{x \in \mathcal{X}} \|\nabla f_t(x) - \nabla F_t(x)\|_2^2\Big],$$

and their cumulative versions $\sigma_{1:T}^2 = \sum_{t=1}^T \sigma_t^2$, $\widetilde{\sigma}_{1:T}^2 = \sum_{t=1}^T \widetilde{\sigma}_t^2$. Always $\sigma_t^2 \le \widetilde{\sigma}_t^2$ by Jensen Inequality, but the gap can be *arbitrarily large*. The following simple 1-dimensional construction (with a growing number of disjoint regions) shows an order difference.

**Proposition A.1** (Linear separation between $\sigma_{1:T}^2$ and $\widetilde{\sigma}_{1:T}^2$). *Let $\mathcal{X} = \bigcup_{i=1}^n X_i \subset \mathbb{R}$ with disjoint cells $X_i = [i-1, i)$ for $i \in \{1, \ldots, n\}$. For each round $t \in \{1, \ldots, T\}$ and each cell $i \in \{1, \ldots, n\}$, draw independently*

$$c_{t,i} \sim \text{Bernoulli}\big(\tfrac{1}{n}\big) \quad and \quad s_{t,i} \in \{-1, +1\} \ \ with \ \ \Pr(s_{t,i} = 1) = \Pr(s_{t,i} = -1) = \tfrac{1}{2},$$

*and define the (stochastic) gradient field by*

$$\nabla f_t(x) = c_{t,i}\, s_{t,i} \qquad for \ all \ x \in X_i,$$

*with arbitrary values on cell boundaries. Let $F_t$ be the expected loss, defined up to an additive constant by $\nabla F_t(x) = \mathbb{E}[\nabla f_t(x)]$ (we fix the constant so that $F_t \equiv 0$). Consider the two gradient–noise proxies*

$$\sigma_t^2 := \sup_{x \in \mathcal{X}} \mathbb{E}\Big[\|\nabla f_t(x) - \nabla F_t(x)\|^2\Big], \qquad \widetilde{\sigma}_t^2 := \mathbb{E}\Big[\sup_{x \in \mathcal{X}} \|\nabla f_t(x) - \nabla F_t(x)\|^2\Big],$$

*and their sums $\sigma_{1:T}^2 = \sum_{t=1}^T \sigma_t^2$ and $\widetilde{\sigma}_{1:T}^2 = \sum_{t=1}^T \widetilde{\sigma}_t^2$.*

*Then, for every $n, T \ge 1$,*

$$\sigma_t^2 = \frac{1}{n} \quad and \quad \widetilde{\sigma}_t^2 = 1 - \Big(1 - \tfrac{1}{n}\Big)^n \xrightarrow[n \to \infty]{} 1 - \tfrac{1}{e},$$

*hence, for all large $n$,*

$$\sigma_{1:T}^2 = \frac{T}{n} = \Theta(1) \quad and \quad \widetilde{\sigma}_{1:T}^2 = T\Big(1 - \Big(1 - \tfrac{1}{n}\Big)^n\Big) = \Theta(T).$$

*In particular, taking $n = n(T) = T$ yields $\sigma_{1:T}^2 = 1$ while $\widetilde{\sigma}_{1:T}^2 \ge (1 - e^{-1})T$, i.e., a linear (in $T$) separation between the two quantities.*

*Proof.* By construction and independence, for any $x \in X_i$,

$$\nabla F_t(x) = \mathbb{E}[\nabla f_t(x)] = \mathbb{E}[c_{t,i}]\,\mathbb{E}[s_{t,i}] = \tfrac{1}{n} \cdot 0 = 0,$$

so $F_t \equiv 0$ (up to an additive constant). Therefore

$$\sigma_t^2 = \sup_{x \in \mathcal{X}} \mathbb{E}\big[(\nabla f_t(x))^2\big] = \sup_i \mathbb{E}[c_{t,i}^2 s_{t,i}^2] = \sup_i \mathbb{E}[c_{t,i}] = \tfrac{1}{n}.$$

Summing over $t$ gives $\sigma_{1:T}^2 = T/n$.

For $\widetilde{\sigma}_t^2$, note that $s_{t,i}^2 \equiv 1$ and $(\nabla f_t(x))^2 = c_{t,i}$ for all $x \in X_i$. Thus

$$\widetilde{\sigma}_t^2 = \mathbb{E}\Big[\sup_{x \in \mathcal{X}}(\nabla f_t(x))^2\Big] = \mathbb{E}\Big[\max_i c_{t,i}\Big] = 1 - \Pr(\forall i, \ c_{t,i} = 0) = 1 - \Big(1 - \tfrac{1}{n}\Big)^n.$$

Since $\big(1 - \tfrac{1}{n}\big)^n \le e^{-1}$ for all $n$, we have $\widetilde{\sigma}_t^2 \in [1 - e^{-1}, 1]$. Summing over $t$ yields $\widetilde{\sigma}_{1:T}^2 = T\big(1 - (1 - \tfrac{1}{n})^n\big) \ge (1 - e^{-1})T$. Finally, with $n = T$ we obtain $\sigma_{1:T}^2 = 1$ and $\widetilde{\sigma}_{1:T}^2 \ge (1 - e^{-1})T$, which proves the claimed linear separation. $\qquad\square$

**Remark.** This example shows that regret bounds expressed in terms of $\widetilde{\sigma}_{1:T}^2 = \sum_t \mathbb{E}[\sup_x \|\cdot\|^2]$ can be a factor $\Theta(T)$ looser than bounds in terms of $\sigma_{1:T}^2 = \sum_t \sup_x \mathbb{E}[\|\cdot\|^2]$ on the same instance.

# B Omitted Details of Section 2

## B.1 Proof of (2)

*Proof.* From Theorem 5 in [27], we have

$$\mathfrak{R}_T(u) \leq \widetilde{O}\left(\|u\|_2 \sqrt{\sum_{t=1}^{T} \|\nabla f_t(x_t) - \nabla f_{t-1}(x_t)\|_2^2}\right)$$

$$\leq \widetilde{O}\left(\|u\|_2 \sqrt{\sum_{t=1}^{T} \sup_{x \in \mathcal{X}} \|\nabla f_t(x) - \nabla f_{t-1}(x)\|_2^2}\right)$$

From Lemma 8 in [12], we also have

$$\sum_{t=1}^{T} \sup_{x \in \mathcal{X}} \|\nabla f_t(x) - \nabla f_{t-1}(x)\|_2^2$$

$$\leq G^2 + 6 \sum_{t=1}^{T} \sup_{x \in \mathcal{X}} \|\nabla f_t(x) - \nabla F_t(x)\|_2^2 + 4 \sum_{t=1}^{T} \sup_{x \in \mathcal{X}} \|\nabla F_t(x) - \nabla F_{t-1}(x)\|_2^2.$$

Taking expectations with Jensen's inequality and the definition of $\widetilde{\sigma}_{1:T}^2$ and $\Sigma_{1:T}^2$, we obtain

$$\mathbb{E}[\mathfrak{R}_T(u)] \leq \widetilde{O}\left(\|u\|_2(\sqrt{\widetilde{\sigma}_{1:T}^2} + \sqrt{\Sigma_{1:T}^2})\right). \qquad \square$$

# C Omitted Details of Section 3

## C.1 Auxiliary Lemmas

**Lemma C.1** (Bregman Proximal Inequality). *The Bregman Proximal update in the form of* $x_{t+1} = \arg\min_{x \in \mathcal{X}} \{\langle x, g_t \rangle + D_\psi(x, x_t)\}$ *satisfies*

$$\langle g_t, x_{t+1} - u \rangle \leq D_\psi(u, x_t) - D_\psi(u, x_{t+1}) - D_\psi(x_t, x_{t+1}). \tag{10}$$

*Proof.* By the first-order optimality condition at $x_{t+1}$, for any $u \in \mathcal{X}$, we have

$$\langle g_t + \nabla \psi(x_{t+1}) - \nabla \psi(x_t), u - x_{t+1} \rangle \geq 0,$$

On the RHS of (10), we expand each term by the definition of Bregman divergence

$$D_\psi(u, x_t) - D_\psi(u, x_{t+1}) - D_\psi(x_t, x_{t+1}) = \langle \nabla \psi(x_{t+1}) - \nabla \psi(x_t), u - x_{t+1} \rangle.$$

Hence, the proof is finished by rearranging the terms. $\qquad \square$

**Lemma C.2.** *Let* $x_t = \arg\min_{x \in \mathcal{X}} \{\langle x, m_t \rangle + D_{\psi_t}(x, x_t')\}$ *and* $x_{t+1}' = \arg\min_{x \in \mathcal{X}} \{\langle x, g_t \rangle + D_{\psi_t}(x, x_t')\}$. *Then, it holds for any* $u$ *in* $\mathcal{X}$

$$\sum_{t=1}^{T} \langle x_t - u, g_t \rangle \leq \sum_{t=1}^{T} \langle x_t - x_{t+1}', g_t - m_t \rangle + D_{\psi_t}(u, x_t') - D_{\psi_t}(u, x_{t+1}')$$

$$- D_{\psi_t}(x_t, x_{t+1}') - D_{\psi_t}(x_t, x_t').$$

*Proof.* We have

$$\langle x_t - u, g_t \rangle = \langle x_t - x_{t+1}', m_t \rangle + \langle x_{t+1}' - u, g_t \rangle + \langle x_t - x_{t+1}', g_t - m_t \rangle. \tag{11}$$

We apply Lemma C.1 twice, i.e., $\langle a - u, f \rangle \leq D_\psi(u, b) - D_\psi(u, a) - D_\psi(a, b)$ since $a = \arg\min_{x \in \mathcal{X}} \langle x, f \rangle + D_\psi(x, b)$. Then, we have

$$\langle x_t - x_{t+1}', m_t \rangle \leq D_{\psi_t}(x_{t+1}', x_t') - D_{\psi_t}(x_{t+1}', x_t) - D_{\psi_t}(x_t, x_t'),$$

$$\langle x_{t+1}' - u, g_t \rangle \leq D_{\psi_t}(u, x_t') - D_{\psi_t}(x_{t+1}', u) - D_{\psi_t}(x_t', x_{t+1}').$$

Substitute these two back to (11) and sum over $T$ providing the desired result. $\qquad \square$

**Lemma C.3.** *In OONS, we have* $0 \leq \langle x_t - x'_{t+1}, \nabla_t - m_t \rangle \leq 2\|\nabla_t - m_t\|^2_{A_t^{-1}}$ *and* $\sum_{t=1}^T \langle x_t - x'_{t+1}, \nabla_t - m_t \rangle \leq O\left( \frac{r \ln(T \eta_1 z_T / z_1)}{\eta_T} \right)$.

*Proof.* We define
$$F_{m_t}(x) = \langle x, m_t \rangle + D_{\psi_t}(x, x'_t), \qquad F_{\nabla_t}(x) = \langle x, \nabla_t \rangle + D_{\psi_t}(x, x'_t)$$
By OONS, we have
$$x_t = \arg\min_{x \in \mathcal{X}} F_{m_t}(x), \qquad x'_{t+1} = \arg\min_{x \in \mathcal{X}} F_{\nabla_t}(x).$$
Since $\nabla^2 D_{\psi_t} = A_t$ and by the first-order optimality at $x'_{t+1}$, we have
$$F_{\nabla_t}(x_t) - F_{\nabla_t}(x'_{t+1}) \geq \frac{1}{2}\|x_t - x'_{t+1}\|^2_{A_t}.$$
Also, we can write $F_{\nabla_t}(x_t) - F_{\nabla_t}(x'_{t+1})$ as
$$F_{\nabla_t}(x_t) - F_{\nabla_t}(x'_{t+1}) = \langle x_t - x'_{t+1}, \nabla_t \rangle + D_{\psi_t}(x_t, x'_t) - D_{\psi_t}(x'_{t+1}, x'_t).$$
Then,
$$\frac{1}{2}\|x_t - x'_{t+1}\|^2_{A_t} \leq \langle x_t - x'_{t+1}, \nabla_t - m_t \rangle + F_{m_t}(x_t) - F_{m_t}(x'_{t+1}),$$
$$\leq \langle x_t - x'_{t+1}, \nabla_t - m_t \rangle,$$
where the second inequality comes from $x_t = \arg\min F_{m_t}(x)$. Therefore, we have
$$\langle x_t - x'_{t+1}, \nabla_t - m_t \rangle \leq 2\|\nabla_t - m_t\|^2_{A_t^{-1}}.$$
Since $x'_{t+1}$ minimizes $F_{m_t}(x)$ and $x_t$ minimizes $F_{\nabla_t}(x)$, we have
$$0 \leq F_{\nabla_t}(x_t) - F_{\nabla_t}(x'_{t+1}) = \langle x_t - x'_{t+1}, \nabla_t \rangle + D_{\psi_t}(x_t, x'_t) - D_{\psi_t}(x'_{t+1}, x'_t),$$
$$= \langle x_t - x'_{t+1}, \nabla_t - m_t \rangle + F_{m_t}(x_t) - F_{m_t}(x'_{t+1})$$
$$\leq \langle x_t - x'_{t+1}, \nabla_t - m_t \rangle.$$
By the definition of $\nabla_t = g_t + 32\eta_t \langle x_t, g_t - m_t \rangle (g_t - m_t)$, we have
$$\|\nabla_t - m_t\|_2 = \|g_t - m_t + 32\eta_t \langle x_t, g_t - m_t \rangle (g_t - m_t)\|_2$$
$$\leq \|g_t - m_t\|_2 + 32\eta_t D \|g_t - m_t\|_2^2 \leq \frac{3}{2}\|g_t - m_t\|_2. \qquad (12)$$
Next, we define
$$\bar{A}_t = 4z_1^2 \cdot I + \sum_{s=1}^t \eta_s (\nabla_s - m_s)(\nabla_s - m_s)^\top.$$
Hence, $A_t \succeq \bar{A}_t$ since $\|\nabla_t - m_t\|_2^2 \leq 4\|g_t - m_t\|_2^2 \leq 4z_t^2$. Also, we have
$$(\nabla_t - m_t)(\nabla_t - m_t)^\top = \frac{1}{\eta_t}[\eta_t(\nabla_t - m_t)(\nabla_t - m_t)^\top] = \frac{1}{\eta_t}(\bar{A}_t - \bar{A}_{t-1})$$
Then,
$$\sum_{t=1}^T \|\nabla_t - m_t\|^2_{A_t^{-1}} \leq \sum_{t=1}^T \|\nabla_t - m_t\|^2_{\bar{A}_t^{-1}},$$
$$= \sum_{t=1}^T \mathrm{trace}\left( (\nabla_t - m_t)(\nabla_t - m_t)^\top \bar{A}_t^{-1} \right),$$
$$\leq \sum_{t=1}^T \frac{1}{\eta_t} \mathrm{trace}\left( \bar{A}_t^{-1}(\bar{A}_t - \bar{A}_{t-1}) \right),$$
$$\leq \sum_{t=1}^T \frac{1}{\eta_t}(\ln|\bar{A}_t| - \ln|\bar{A}_{t-1}|),$$
$$\leq \frac{1}{\eta_T} \ln \frac{|\bar{A}_T|}{|\bar{A}_0|}.$$

For $|\bar{A}_T|$:

$$|\bar{A}_T| \le |4z_1^2 I + \sum_{t=1}^{T} \eta_t (\nabla_t - m_t)(\nabla_t - m_t)^{\top}|$$

$$\ln|\bar{A}_T| \le O\Big(r\ln(1 + \sum_{t=1}^{T} \frac{\eta_t}{4z_1^2}\|\nabla_t - m_t\|_2^2)\Big)$$

$$\le O\Big(r\ln(1 + \frac{4z_T^2}{4z_1^2}\sum_{t=1}^{T}\eta_t)\Big)$$

$$\le O\Big(r\ln(1 + \frac{\eta_1 z_T^2}{z_1^2}T)\Big),$$

where $r$ is the rank of $\sum_{t=1}^{T}(\nabla_t - m_t)(\nabla_t - m_t)^{\top}$.

Therefore, we have

$$\sum_{t=1}^{T}\langle x_t - x'_{t+1}, \nabla_t - m_t\rangle \le O\Big(\frac{r\ln(T\eta_1 z_T/z_1)}{\eta_T}\Big).$$

$\square$

**Lemma C.4.** *Let $s_t, \forall t \in [T]$ be non-negative. Then,*

$$\sum_{t=1}^{T} \frac{s_t}{\sqrt{\sum_{j=1}^{t} s_j}} \le 2\sqrt{\sum_{t=1}^{T} s_t}.$$

*Proof.* Let $S_t = \sum_{j=1}^{t} s_j$. Then,

$$\sum_{t=1}^{T} \frac{s_t}{\sqrt{S_t}} \le \sum_{t=1}^{T} \int_{S_{t-1}}^{S_t} \frac{1}{\sqrt{x}}dx = \int_{0}^{S_T} \frac{1}{\sqrt{x}}dx = 2\sqrt{S_T}.$$

$\square$

**Lemma C.5** (Theorem 5 in [8], Lemma 3 in [12])**.** *Under Assumptions 2.1 and 2.3, we have*

$$\sum_{t=1}^{T}\|\nabla f_t(x_t) - \nabla f_{t-1}(x_{t-1})\|_2^2 \le G^2 + 4\sum_{t=2}^{T}\|\nabla F_t(x_{t-1}) - \nabla F_{t-1}(x_{t-1})\|_2^2$$

$$+ 8\sum_{t=1}^{T}\|\nabla f_t(x_t) - \nabla F_t(x_t)\|_2^2 + 4L^2\sum_{t=2}^{T}\|x_t - x_{t-1}\|_2^2.$$

## C.2 Proof of Theorem 3.1

*Proof.* By Lemma C.1, we have

$$\sum_{t=1}^{T}\langle x_t - u, \nabla_t\rangle$$

$$\le \sum_{t=1}^{T}\langle x_t - x'_{t+1}, \nabla_t - m_t\rangle + D_{\psi_t}(u, x'_t) - D_{\psi_t}(u, x'_{t+1}) - D_{\psi_t}(x_t, x'_{t+1}) - D_{\psi_t}(x_t, x'_t),$$

$$\le \sum_{t=1}^{T}\langle x_t - x'_{t+1}, \nabla_t - m_t\rangle + D_{\psi_1}(u, x'_1) + \sum_{t=1}^{T-1} D_{\psi_{t+1}}(u, x'_{t+1}) - D_{\psi_t}(u, x'_{t+1})$$

$$- \sum_{t=1}^{T}(D_{\psi_t}(x_t, x'_{t+1}) + D_{\psi_t}(x_t, x'_t))$$

**Term** $D_{\psi_1}(u, x_1')$: Since the initialization of $x_1' = 0$ and $A_1 = O(z_1^2 I)$, we have

$$D_{\psi_1}(u, x_1') = \frac{1}{2}\|u\|_{A_1}^2 \leq O(z_1^2 \|u\|_2^2).$$

**Term** $\sum_{t=1}^{T-1} D_{\psi_{t+1}}(u, x_{t+1}') - D_{\psi_t}(u, x_{t+1}')$: First, we have

$$A_{t+1} - A_t = \eta_t(\nabla_t - m_t)(\nabla_t - m_t)^\top + 4\eta_t(z_{t+1}^2 - z_t^2)I.$$

Also,

$$D_{\psi_{t+1}}(u, x_{t+1}') - D_{\psi_t}(u, x_{t+1}') = \frac{1}{2}(u - x_{t+1}')^\top (A_{t+1} - A_t)(u - x_{t+1}')$$

$$= \frac{1}{2}(u - x_{t+1}')^\top (\eta_t(\nabla_t - m_t)(\nabla_t - m_t)^\top + 4\eta_t(z_{t+1}^2 - z_t^2)I)(u - x_{t+1}')$$

$$= \frac{\eta_t}{2}\left\langle u - x_{t+1}', \nabla_t - m_t\right\rangle^2 + 2\eta_t(z_{t+1}^2 - z_t^2)\|u - x_{t+1}'\|_2^2.$$

Then, we have

$$\sum_{t=1}^{T-1} D_{\psi_{t+1}}(u, x_{t+1}') - D_{\psi_t}(u, x_{t+1}')$$

$$\leq \sum_{t=1}^{T-1} \frac{\eta_t}{2}\langle u - x_{t+1}', \nabla_t - m_t\rangle^2 + O\Big(\sum_{t=1}^{T-1}\eta_t D^2(z_{t+1}^2 - z_t^2)\Big),$$

$$\leq \sum_{t=1}^{T-1} \frac{\eta_t}{2}\langle u - x_{t+1}', \nabla_t - m_t\rangle^2 + O\Big(\sum_{t=1}^{T-1} D(z_{t+1}^2 - z_t^2)/z_T\Big), \qquad \text{(since } \eta_t \leq \frac{1}{64Dz_T}\text{)}$$

$$\leq \sum_{t=1}^{T-1} \frac{\eta_t}{2}\langle u - x_{t+1}', \nabla_t - m_t\rangle^2 + O(D(z_T - z_1)),$$

$$\leq \sum_{t=1}^{T-1} \eta_t\langle u - x_t, \nabla_t - m_t\rangle^2 + \sum_{t=1}^{T-1} \eta_t\langle x_t - x_{t+1}', \nabla_t - m_t\rangle^2 + O(D(z_T - z_1)),$$

$$\leq \sum_{t=1}^{T-1} \eta_t\langle u - x_t, \nabla_t - m_t\rangle^2 + \sum_{t=1}^{T-1} \eta_t\langle x_t - x_{t+1}', \nabla_t - m_t\rangle\langle x_t - x_{t+1}', \nabla_t - m_t\rangle$$

$$+ O(D(z_T - z_1)),$$

$$\leq \sum_{t=1}^{T-1} \eta_t\langle u - x_t, \nabla_t - m_t\rangle^2 + \sum_{t=1}^{T-1} 2\eta_t Dz_t\langle x_t - x_{t+1}', \nabla_t - m_t\rangle + O(D(z_T - z_1)),$$

$$\text{(Using Equation 12)}$$

$$\leq \sum_{t=1}^{T-1} \eta_t\langle u - x_t, \nabla_t - m_t\rangle^2 + O\Big(\frac{r\ln(T\eta_1 z_T/z_1)}{\eta_T} + D(z_T - z_1)\Big),$$

$$\text{(Using Lemma C.3)}$$

$$\leq \sum_{t=1}^{T-1} 2\eta_t\langle u, \nabla_t - m_t\rangle^2 + 2\eta_t\langle x_t, \nabla_t - m_t\rangle^2 + O\Big(\frac{r\ln(T\eta_1 z_T/z_1)}{\eta_T} + D(z_T - z_1)\Big),$$

$$\leq \sum_{t=1}^{T-1} 8\eta_t\langle u, g_t - m_t\rangle^2 + 8\eta_t\langle x_t, g_t - m_t\rangle^2 + O\Big(\frac{r\ln(T\eta_1 z_T/z_1)}{\eta_T} + D(z_T - z_1)\Big).$$

$$\text{(since } 32\eta_t\langle x_t, g_t - m_t\rangle \leq \frac{1}{2}\text{)}$$

**Term** $\sum_{t=1}^T (D_{\psi_t}(x_t, x'_{t+1}) + D_{\psi_t}(x_t, x'_t))$:

$$\sum_{t=1}^T (D_{\psi_t}(x_t, x'_{t+1}) + D_{\psi_t}(x_t, x'_t)) = \sum_{t=1}^T \frac{1}{2}(\|x_t - x'_t\|_{A_t}^2 + \|x'_{t+1} - x_t\|_{A_t}^2)$$

$$\geq \frac{1}{2} \sum_{t=2}^T \|x_t - x'_t\|_{A_{t-1}}^2 + \frac{1}{2} \sum_{t=2}^{T+1} \|x_{t-1} - x'_t\|_{A_{t-1}}^2$$

$$\geq \frac{4z_1^2}{4} \sum_{t=2}^T \|x_t - x_{t-1}\|_2^2 = z_1^2 \sum_{t=2}^T \|x_t - x_{t-1}\|_2^2,$$

where the last inequality comes from $A_t \succeq A_{t-1} \succeq 4z_1^2 I$.

Since $c_t(x)$ is convex in $x$, we have

$$\sum_{t=1}^T c_t(x_t) - c_t(u)$$

$$= \sum_{t=1}^T \langle x_t - u, g_t \rangle + 16\eta_t \langle x_t, g_t - m_t \rangle^2 - 16\eta_t \langle u, g_t - m_t \rangle^2 \leq \sum_{t=1}^T \langle \nabla_t, x_t - u \rangle.$$

Therefore, the final regret bound here is

$$\mathfrak{R}_T(u) \leq \sum_{t=1}^T \langle x_t - u, g_t \rangle$$

$$\leq O\left( \frac{r \ln(T\eta_1 z_T/z_1)}{\eta_T} + z_1^2 \|u\|_2^2 + D(z_T - z_1) + \sum_{t=1}^T \eta_t \langle u, g_t - m_t \rangle^2 - z_1^2 \sum_{t=2}^T \|x_t - x_{t-1}\|_2^2 \right).$$

$\square$

### C.3 Proof of Theorem 3.2

*Proof.* In this case, we have $\|g_t - m_t\|_2 \leq 2G, \forall t \in [T]$. Then, we set $z_t = 2G, \forall t \in [T]$ and step-size $\eta_t$ as

$$\eta_t = \min\left\{ \frac{1}{64Dz_T}, \frac{1}{D\sqrt{\sum_{s=1}^{t-1} \|g_s - m_s\|_2^2}} \right\} \quad \text{for all} \quad t \in [T].$$

By substituting $\eta_t$ and $z_t$ into (3), we have

$$\mathfrak{R}_T(u) \leq \widetilde{O}\left( D\sqrt{\sum_{t=1}^{T-1} \|g_t - m_t\|_2^2} + G^2 D^2 + \sum_{t=1}^T \eta_t \langle u, g_t - m_t \rangle^2 - G^2 \sum_{t=2}^T \|x_t - x_{t-1}\|_2^2 \right).$$

By Lemma C.4, we have

$$\sum_{t=1}^T \eta_t \langle u, g_t - m_t \rangle^2 \leq 2D\sqrt{\sum_{t=1}^T \|g_t - m_t\|_2^2}$$

Then,

$$\mathfrak{R}_T(u) \leq \widetilde{O}\left( D\sqrt{\sum_{t=1}^T \|g_t - m_t\|_2^2 - G^2 \|x_t - x_{t-1}\|_2^2} \right).$$

By Lemma C.5 and the definition of $g_t$ and $m_t$, we have

$$D\sqrt{\sum_{t=1}^{T}\|g_t - m_t\|_2^2 - G^2\sum_{t=2}^{T}\|x_t - x_{t-1}\|_2^2}$$

$$\leq DG + 2D\sqrt{\sum_{t=2}^{T}\|\nabla F_t(x_{t-1}) - \nabla F_{t-1}(x_{t-1})\|_2^2} + 2\sqrt{2}D\sqrt{\sum_{t=1}^{T}\|\nabla f_t(x_t) - \nabla F_t(x_t)\|_2^2}$$

$$+ 2LD\sqrt{\sum_{t=2}^{T}\|x_t - x_{t-1}\|_2^2 - G^2\sum_{t=2}^{T}\|x_t - x_{t-1}\|_2^2}$$

$$\leq DG + \frac{D^2L^2}{G^2} + 2D\sqrt{\sum_{t=2}^{T}\|\nabla F_t(x_{t-1}) - \nabla F_{t-1}(x_{t-1})\|_2^2}$$

$$+ 2\sqrt{2}D\sqrt{\sum_{t=1}^{T}\|\nabla f_t(x_t) - \nabla F_t(x_t)\|_2^2}$$

Therefore, we have

$$\mathbb{E}[\mathfrak{R}_T(u)] \leq \widetilde{O}(\sqrt{\sigma_{1:T}^2} + \sqrt{\Sigma_{1:T}^2})$$

$\square$

## C.4 Computational Complexity of OONS

The primary computational bottleneck in our proposed OONS algorithm (Algorithm 1) is the management of the $d \times d$ matrix $A_t$. A naive implementation would involve storing this dense matrix and performing a full matrix inversion at each step, leading to prohibitive costs in high-dimensional settings.

- **Storage:** Storing the dense $d \times d$ matrix $A_t$ requires $O(d^2)$ memory.

- **Computation:** A naive matrix inversion $A_t^{-1}$ would cost $O(d^3)$ per step, and the subsequent matrix-vector products would cost $O(d^2)$.

In practice, this complexity can be significantly reduced. Since the matrix $A_t$ is constructed by a sum of outer products ($A_t = cI + \sum \eta_s v_s v_s^\top$), its inverse can be efficiently computed and updated at each step using the Sherman-Morrison-Woodbury formula. This reduces the update complexity from $O(d^3)$ to $O(d^2)$ per step.

However, an $O(d^2)$ complexity per step can still be prohibitive in high-dimensional scenarios. To address this, existing research has explored several techniques:

- **Matrix Sketching:** This technique approximates the original $d \times d$ matrix $A_t$ with a much smaller "sketched" matrix, thereby significantly reducing both storage and computational requirements. For instance, Luo et al. [45] have successfully applied sketching to the Online Newton Step (ONS) algorithm, creating matrix-free updates that avoid direct manipulation of the high-dimensional matrix.

- **Sparsity:** If the gradient vectors are sparse across most iterations, specialized sparse data structures and algorithms can be utilized. This allows update computations to be performed much more efficiently, avoiding the full $O(d^2)$ cost of dense matrix-vector multiplications.

Integrating these high-dimensional adaptation techniques into our proposed algorithms for the SEA model and analyzing their theoretical guarantees is an interesting direction for future work.

---

**Algorithm 5** Multi-scale Multiplicative-weight with Correction (MsMwC)

---

**Input:** $w_1' \in \Delta_N$.

1: **for** $t = 1, \ldots, T$ **do**
2:     Receive the prediction $h_t \in \mathbb{R}^N$.
3:     Compute $w_t = \arg\min_{w \in \Delta_N} \langle w, h_t \rangle + D_\phi(w, w_t')$, where $\phi_t(w) = \sum_{j=1}^N \frac{w_j}{\beta_{t,j}} \ln w_j$.
4:     Play $w_t$, receive $\ell_t$ and construct correction term $a_t \in \mathbb{R}^N$ with $a_{t,j} = 32\beta_{t,j}(\ell_t^j - m_t^j)^2$.
5:     Compute $w_{t+1}' = \arg\min_{w \in \Delta_N} \langle w, \ell_t + a_t \rangle + D_\phi(w, w_t')$.
6: **end for**

---

## D    Omitted Details of Section 4.1

### D.1    Multi-scale Multiplicative-weight with Correction (MsMwC)

We rephrase the MsMwC algorithm [10] as the following Algorithm 5.

### D.2    Proof of equation (5)

The final decision $x_t$ is a weighted-average of base-learners' decisions: $x_t = \sum_{j=1}^N w_{t,j} x_t^j$. Then,

$$
\begin{aligned}
\mathfrak{R}_T(u) &= \mathfrak{R}_T^{\mathcal{A}_i}(u) + \sum_{t=1}^T f_t(x_t) - \sum_{t=1}^T f_t(x_t^i) \\
&= \mathfrak{R}_T^{\mathcal{A}_i}(u) + \sum_{t=1}^T f_t\left(\sum_{j \in [N]} w_{t,j} x_t^j\right) - \sum_{t=1}^T f_t(x_t^i) \\
&\leq \mathfrak{R}_T^{\mathcal{A}_i}(u) + \sum_{t=1}^T \sum_{j \in [N]} w_{t,j} f_t(x_t^j) - \sum_{t=1}^T f_t(x_t^i) \\
&= \mathfrak{R}_T^{\mathcal{A}_i}(u) + \sum_{t=1}^T \sum_{j \in [N]} f_t(x_t^j)[w_{t,j} - \mathbb{1}(j = i)] \\
&= \mathfrak{R}_T^{\mathcal{A}_i}(u) + \sum_{t=1}^T \sum_{j \in [N]} (f_t(x_t^j) - f_t(0))[w_{t,j} - \mathbb{1}(j = i)] \\
&\leq \mathfrak{R}_T^{\mathcal{A}_i}(u) + \sum_{t=1}^T \sum_{j \in [N]} \langle \nabla f_t(x_t^j), x_t^j \rangle [w_{t,j} - \mathbb{1}(j = i)] \\
&= \mathfrak{R}_T^{\mathcal{A}_i}(u) + \sum_{t=1}^T \langle \ell_t, w_t - w_\star^i \rangle,
\end{aligned}
$$

where $\ell_t \in \mathbb{R}^N$ with $\ell_t^j = \langle \nabla f_t(x_t^j), x_t^j \rangle$ and $w_\star^i$ is a vector in $\Delta_N$ whose $j$-th component is $(w_\star^i)_j = 1$ if $j = i$ and 0 otherwise.

### D.3    Auxiliary Lemma

**Lemma D.1.** *(Theorem 6 in [10]) Suppose for all $t \in [T]$ and $j \in [N]$, $|\ell_t^j| \leq GD_j$ and $|h_t^j| \leq GD_j$, where $\ell_t^j = \langle \nabla f_t(x_t^j), x_t^j \rangle$ and $h_t^j = \langle \nabla f_{t-1}(x_{t-1}^j), x_t^j \rangle$. Define $\Gamma_j = \ln(\frac{NTD_j}{D_1})$ and the set $\mathcal{E}$ as*

$$
\mathcal{E} = \left\{ (\beta_k, \mathcal{G}_k) : \forall k \in \mathcal{S}, \beta_k = \frac{1}{32 \cdot 2^k} \right\},
$$

*where $\mathcal{G}_k$ is the MsMwC algorithm with $w_1'$ being uniform over $\mathcal{Z}(k)$, $\mathcal{S} = \{k \in \mathbb{Z} : \exists j \in [N], GD_j \leq 2^{k-2} \leq GD_j\sqrt{T}\}$ and $\mathcal{Z}_k = \{j \in [N] : GD_j \leq 2^{k-2}\}$. We have the following regret*

*bound*

$$\sum_{t=1}^{T}\langle \ell_t, w_t - w_\star^i\rangle \leq O\Big(D_i\Gamma_i + \sqrt{\Gamma_i \sum_{t=1}^{T}(\ell_t^i - h_t^i)^2}\Big).$$

*Proof.* The regret $\sum_{t=1}^{T}\langle \ell_t, w_t - w_\star^i\rangle$ can also be decomposed as

$$\sum_{t=1}^{T}\langle \ell_t, w_t - w_\star^i\rangle = \sum_{t=1}^{T}\Big\langle \ell_t, w_t^{k_\star} - w_\star^i\Big\rangle + \sum_{t=1}^{T}\Big\langle \ell_t, w_t - w_t^{k_\star}\Big\rangle$$

$$= \sum_{t=1}^{T}\Big\langle \ell_t, w_t^{k_\star} - w_\star^i\Big\rangle + \sum_{t=1}^{T}\langle L_t, p_t - e_{k_\star}\rangle,$$

where $e_{k_\star}$ is the $k_\star$-th standard basis vector and the second equality is from the definition of $L_t$.

For any $i \in [N]$, there exists a $k_\star$ such that $\eta_{k_\star} \leq \min\Big\{\frac{1}{128GD_i}, \sqrt{\frac{\Gamma_i}{\sum_{t=1}^{T}(\ell_t^i - h_t^i)^2}}\Big\} \leq 2\eta_{k_\star}$. By Lemma 1 and Theorem 4 in [10], we have

$$\sum_{t=1}^{T}\langle \ell_t, w_t - w_\star^i\rangle \leq O\Big(D_i\Gamma_i + \sqrt{\Gamma_i \sum_{t=1}^{T}(\ell_t^i - h_t^i)^2}\Big).$$

$\square$

### D.4 Proof of Theorem 4.1

*Proof.* We begin with considering the first and second cases that $\|u\|_2 \leq D_1$ and $\|u\|_2 \leq D_i \leq 2\|u\|_2$.

Here, we define $g_t^j = \nabla f_t(x_t^j)$ and $m_t^j = \nabla f_{t-1}(x_{t-1}^j)$ for all $t \in [T]$ and $j \in [N]$. For the meta regret, we define $\ell_t \in \mathbb{R}^N$ with $\ell_t^j = \langle \nabla f_t(x_t^j), x_t^j\rangle$ and $h_t \in \mathbb{R}^N$ with $h_t^j = \langle \nabla f_{t-1}(x_{t-1}^j), x_t^j\rangle$. Then, $|\ell_t^j| \leq GD_j$ and $|h_t^j| \leq GD_j$ for all $t \in [T]$ and $j \in [N]$. By applying Lemma D.1, we have

$$\sum_{t=1}^{T}\langle \ell_t, w_t - w_\star^i\rangle \leq O\Big(D_i\Gamma_i + \sqrt{\Gamma_i \sum_{t=1}^{T}(\ell_t^i - h_t^i)^2}\Big).$$

By the definition of $\ell_t^i$ and $h_t^i$, we have

$$\sum_{t=1}^{T}(\ell_t^i - h_t^i)^2 \leq \sum_{t=1}^{T}\langle \nabla f_t(x_t^i) - \nabla f_{t-1}(x_{t-1}^i), x_t^i\rangle^2$$

$$\leq D_i^2 \sum_{t=1}^{T}\|\nabla f_t(x_t^i) - \nabla f_{t-1}(x_{t-1}^i)\|_2^2 = D_i^2 \sum_{t=1}^{T}\|g_t^i - m_t^i\|_2^2.$$

Hence,

$$\sum_{t=1}^{T}\langle \ell_t, w_t - w_\star^i\rangle \leq O\Big(D_i\Gamma_i + D_i\sqrt{\Gamma_i \sum_{t=1}^{T}\|g_t^i - m_t^i\|_2^2}\Big). \tag{13}$$

Then, we investigate the expert regret part. Here, we set the step-size for the expert $i$ as

$$\eta_t^i = \min\Big\{\frac{1}{64D_i z_T}, \frac{1}{D_i\sqrt{\sum_{s=1}^{t-1}\|g_s^i - m_s^i\|_2^2}}\Big\}. \tag{14}$$

By substituting the step-size specified in (14) to (3), we have

$$\mathfrak{R}_T^{\mathcal{A}_i}(u) \leq \widetilde{O}\Big(D_i\sqrt{\sum_{t=1}^{T}\|g_t^i - m_t^i\|_2^2 + G^2\|u\|_2^2} + \frac{\|u\|_2^2}{D_i}\sqrt{\sum_{t=1}^{T}\|g_t^i - m_t^i\|_2^2 - G^2\|x_t^i - x_{t-1}^i\|_2^2}\Big)$$

$$\leq \widetilde{O}\Big(D_i^2 + D_i\sqrt{\sum_{t=1}^{T}\|g_t^i - m_t^i\|_2^2 - G^2\|x_t^i - x_{t-1}^i\|_2^2}\Big). \tag{15}$$

Therefore, by combining (13) and (15), we have

$$\mathfrak{R}_T(u) \leq \widetilde{O}\Big(D_i^2 + D_i\sqrt{\sum_{t=1}^{T}\|g_t^i - m_t^i\|_2^2 - G^2\sum_{t=2}^{T}\|x_t^i - x_{t-1}^i\|_2^2}\Big)$$

Then, by applying Lemma C.5, we have

$$D_i\sqrt{\sum_{t=1}^{T}\|g_t^i - m_t^i\|_2^2 - G^2\sum_{t=2}^{T}\|x_t^i - x_{t-1}^i\|_2^2}$$

$$\leq GD_i + \frac{D_i^2 L^2}{G^2} + 2D_i\sqrt{\sum_{t=2}^{T}\|\nabla F_t(x_{t-1}^i) - \nabla F_{t-1}(x_{t-1}^i)\|_2^2}$$

$$+ 2\sqrt{2}D_i\sqrt{\sum_{t=1}^{T}\|\nabla f_t(x_t^i) - \nabla F_t(x_t^i)\|_2^2}.$$

Since the expert $i$ runs OONS within the set $\mathcal{X}_i = \{x : \|x\|_2 \leq D_i\} \subseteq \mathcal{X}$, we have

$$\sup_{x \in \mathcal{X}_i} \mathbb{E}_{f_t \sim \mathcal{D}_t}\big[\|\nabla f_t(x) - \nabla F_t(x)\|_2^2\big] \leq \sup_{x \in \mathcal{X}} \mathbb{E}_{f_t \sim \mathcal{D}_t}\big[\|\nabla f_t(x) - \nabla F_t(x)\|_2^2\big]$$

$$\sup_{x \in \mathcal{X}_i} \|\nabla F_t(x) - \nabla F_{t-1}(x)\|_2^2 \leq \sup_{x \in \mathcal{X}} \|\nabla F_t(x) - \nabla F_{t-1}(x)\|_2^2.$$

Hence,

$$\mathbb{E}[\mathfrak{R}_T(u)] \leq \widetilde{O}\Big(D_i^2 + \frac{D_i^2 L^2}{G^2} + D_i\sqrt{\sigma_{1:T}^2} + D_i\sqrt{\Sigma_{1:T}^2}\Big). \tag{16}$$

**Case 1**: $\|u\|_2 \leq D_1$. We take $i = 1$ and substitute $D_i$ with $D_1$ into (16). Hence, we have

$$\mathbb{E}[\mathfrak{R}_T(u)] \leq \widetilde{O}\Big(\sqrt{\sigma_{1:T}^2} + \sqrt{\Sigma_{1:T}^2}\Big).$$

**Case 2**: In this case, let $i$ be the smallest integer such that $\|u\|_2 \leq D_i = 2^i$. We have $\|u\|_2 \leq D_i \leq 2\|u\|_2$ since $D_{i+1} = 2D_i$. Then, we substitute $D_i$ with $2\|u\|_2$ into the regret bound (16). Then,

$$\mathbb{E}[\mathfrak{R}_T(u)] \leq \widetilde{O}\Big(\|u\|_2^2 + \|u\|_2(\sqrt{\sigma_{1:T}^2} + \sqrt{\Sigma_{1:T}^2})\Big).$$

**Case 3**: $\|u\|_2 > D_{\max}$.

Next, we consider the case when $\|u\|_2 > D_{\max} = 2^N$. Then, we have

$$\sum_{t=1}^{T} f_t(x_t) - f_t(u) \leq \sum_{t=1}^{T} \langle \nabla f_t(x_t), x_t - u \rangle$$

$$\leq 2GT\|u\|_2.$$

We take $N = \lceil \log T \rceil$, then $T \leq \|u\|_2$. Therefore,

$$\sum_{t=1}^{T} f_t(x_t) - f_t(u) \leq 2G\|u\|_2^2.$$

By combining these two cases above, the desirable regret bound is achieved. $\qquad\square$

# E  Omitted Details of Section 4.2

In this section, we also denote $g_t = \nabla f_t(x_t)$ and $m_t = \nabla f_{t-1}(x_{t-1})$. We first note that

$$\max_{t \leq T} D_t < \sqrt{\sum_{s=1}^{t} \frac{\|g_s\|_2}{\max\{1, \max_{k \leq s} \|g_k\|_2\}}} \leq \sqrt{T} .$$

Thus, we need to update $D_t$ at most $O(\log T)$ times. Let $M$ be the number of total updates in $D_t$, where $M = \mathcal{O}(\log T)$. We split the $T$ iterations into $M$ intervals $I_m$ with $m \in [M]$, where the last iteration of $I_m$ (denoted by $t_m$) either equals to $T$ or $D_{t_m+1} \neq D_{t_m}$.

## E.1  Proof of Theorem 4.5

*Proof.* We have

$$\mathfrak{R}_T(u) = \sum_{t=1}^{T} f_t(x_t) - \sum_{t=1}^{T} f_t(u) \leq \sum_{t=1}^{T} \langle g_t, x_t - u \rangle$$

$$= \sum_{m=1}^{M} \underbrace{\sum_{t \in I_m} \langle g_t, x_t - u_t \rangle}_{T_m} + \underbrace{\sum_{t=1}^{T} \langle g_t, u_t - u \rangle}_{T_{\text{extra}}},$$

where we define $u_t = \min\{1, \frac{D_t}{\|u\|_2}\}u$.

We first consider $T_m$ as

$$\sum_{t \in I_m} \langle g_t, x_t - u_t \rangle = \sum_{t \in I_m} \langle \widetilde{g}_t, x_t - u_t \rangle + \sum_{t \in I_m} \langle g_t - \widetilde{g}_t, x_t - u_t \rangle .$$

Note that iteration $t$ within interval $I_m$, i.e., $t \in I_m$, the domain has a bounded diameter $D_t$. When $t \in I_m$, we take

$$\eta_t = \min\{\frac{1}{64 D_t z_t}, \frac{1}{\sqrt{\sum_{s=t_1}^{t-1} \|\widetilde{g}_s - m_s\|_2^2}}\},$$

where $t_1$ is first index in $I_m$. Also, we denote the last index in $I_m$ as $t_m$, respectively. From the Line 5 or 8 in FPF-OONS we need to reset $x'_{t_1} = 0$ and $A_t$ for all $t \in I_m$ at iteration $t_1$ as follows

$$A_t = 4 z_{t_1}^2 I + \sum_{s=t_1}^{t-1} \eta_s (\nabla_s - m_s)(\nabla_s - m_s)^\top + 4 \eta_t z_t^2 I.$$

Similar to the proof of Theorem 3.1, we have

$$\sum_{t=t_1}^{t_m} \langle x_t - u_t, \nabla_t \rangle$$

$$\leq \sum_{t=t_1}^{t_m} \langle x_t - x'_{t+1}, \nabla_t - m_t \rangle + D_{\psi_{t_1}}(u, x'_{t_1}) + \sum_{t=t_1}^{t_m-1} D_{\psi_{t+1}}(u_t, x'_{t+1}) - D_{\psi_t}(u_t, x'_{t+1})$$

$$- \sum_{t=t_1}^{t_m} (D_{\psi_t}(x_t, x'_{t+1}) + D_{\psi_t}(x_t, x'_t)),$$

where $\nabla_t = \widetilde{g}_t + 32 \eta_t \langle x_t, \widetilde{g}_t - m_t \rangle (\widetilde{g}_t - m_t)$.

We first consider the term $\sum_{t=t_1}^{t_m} \langle x_t - x'_{t+1}, \nabla_t - m_t \rangle$. By Lemma C.3, we have $0 \leq \langle x_t - x'_{t+1}, \nabla_t - m_t \rangle \leq 2\|\nabla_t - m_t\|_{A_t^{-1}}^2$.

Also, we have

$$\|\widetilde{g}_t - m_t\|_2 = \|m_t + \frac{B_{t-1}}{B_t}(g_t - m_t) - m_t\|_2$$

$$\leq \|g_t - m_t\|_2$$

By the definition of $\nabla_t = \widetilde{g}_t + 32\eta_t \langle x_t, \widetilde{g}_t - m_t \rangle (\widetilde{g}_t - m_t)$ and $\|\widetilde{g}_t - m_t\|_2 \leq \|g_t - m_t\|_2$, we have

$$\|\nabla_t - m_t\|_2 = \|\widetilde{g}_t - m_t + 32\eta_t \langle x_t, \widetilde{g}_t - m_t \rangle (\widetilde{g}_t - m_t)\|_2$$

$$\leq \|\widetilde{g}_t - m_t\|_2 + 32\eta_t D_t \|\widetilde{g}_t - m_t\|_2^2$$

$$\leq \|\widetilde{g}_t - m_t\|_2 + 32\eta_t D_t z_t \|\widetilde{g}_t - m_t\|_2$$

$$\leq \frac{3}{2}\|\widetilde{g}_t - m_t\|_2 \leq \frac{3}{2}\|g_t - m_t\|_2.$$

For $t \in I_m$, we also redefine

$$\bar{A}_t = 4z_{t_1}^2 \cdot I + \sum_{s=t_1}^{t} \eta_s (\nabla_s - m_s)(\nabla_s - m_s)^\top.$$

Hence, $A_t \succeq \bar{A}_t$ since $\|\nabla_t - m_t\|_2^2 \leq 4\|\widetilde{g}_t - m_t\|_2^2 \leq 4z_t^2$. Also, for $t \in [t_1 + 1, t_m]$, we have

$$(\nabla_t - m_t)(\nabla_t - m_t)^\top = \frac{1}{\eta_t}[\eta_t(\nabla_t - m_t)(\nabla_t - m_t)^\top] = \frac{1}{\eta_t}(\bar{A}_t - \bar{A}_{t-1})$$

Then,

$$\sum_{t=t_1}^{t_m} \|\nabla_t - m_t\|_{A_t^{-1}}^2 = \|\nabla_{t_1} - m_{t_1}\|_{A_{t_1}^{-1}}^2 + \sum_{t=t_1+1}^{t_m} \|\nabla_t - m_t\|_{A_t^{-1}}^2$$

$$= \frac{1}{4z_{t_1}^2}\|\nabla_{t_1} - m_{t_1}\|_2^2 + \sum_{t=t_1+1}^{t_m} \|\nabla_t - m_t\|_{A_t^{-1}}^2$$

$$\leq 1 + \sum_{t=t_1+1}^{t_m} \|\nabla_t - m_t\|_{A_t^{-1}}^2 \qquad (\text{since } \|\nabla_{t_1} - m_{t_1}\|_2^2 \leq 4z_{t_1}^2)$$

$$\leq \sum_{t=t_1+1}^{t_m} \|\nabla_t - m_t\|_{\bar{A}_t^{-1}}^2 + 1$$

$$\leq \sum_{t=t_1+1}^{t_m} \frac{1}{\eta_t}(\ln|\bar{A}_t| - \ln|\bar{A}_{t-1}|) + 1$$

$$\leq \frac{1}{\eta_{t_m}} \ln \frac{|\bar{A}_{t_m}|}{|\bar{A}_{t_1}|} + 1.$$

For $|\bar{A}_{t_m}|$:

$$|\bar{A}_{t_m}| \leq |4z_{t_1}^2 I + \sum_{t=t_1}^{t_m} \eta_t (\nabla_t - m_t)(\nabla_t - m_t)^\top|.$$

$$\ln|\bar{A}_{t_m}| \leq O\Big(r \ln(1 + \sum_{t=t_1}^{t_m} \frac{\eta_t}{4z_{t_1}^2}\|\nabla_t - m_t\|_2^2)\Big)$$

$$\leq O\Big(r \ln(1 + \frac{4z_{t_m}^2}{4z_{t_1}^2}\sum_{t=t_1}^{t_m} \eta_t)\Big)$$

$$\leq O\Big(r \ln(1 + \frac{\eta_{t_1} z_{t_m}^2}{z_{t_1}^2}T)\Big).$$

Therefore, we have

$$\sum_{t=t_1}^{t_m} \langle x_t - x'_{t+1}, \nabla_t - m_t \rangle \le O\Big(\frac{r \ln(T\eta_{t_1} z_{t_m}/z_{t_1})}{\eta_{t_m}}\Big).$$

**Term** $D_{\psi_{t_1}}(u, x'_{t_1})$:

$$D_{\psi_{t_1}}(u_t, x'_{t_1}) = \frac{1}{2}\|u_t\|_{A_{t_1}}^2 \le O(z_{t_1}^2 \|u_t\|_2^2).$$

**Term** $\sum_{t=t_1}^{t_m-1} D_{\psi_{t+1}}(u_t, x'_{t+1}) - D_{\psi_t}(u_t, x'_{t+1})$:

By the definition of $u_t = \min\{1, \frac{D_t}{\|u\|_2}\}u$, we have

$$\|u_t\|_2 = \min\{\|u\|_2, D_t\}.$$

Then, we have

$$\sum_{t=t_1}^{t_m-1} D_{\psi_{t+1}}(u_t, x'_{t+1}) - D_{\psi_t}(u_t, x'_{t+1})$$

$$\le \sum_{t=t_1}^{t_m-1} \frac{\eta_t}{2} \left\langle u_t - x'_{t+1}, \nabla_t - m_t \right\rangle^2 + O\Big(\sum_{t=t_1}^{t_m-1} \eta_t \|u_t - x'_{t+1}\|_2^2 (z_{t+1}^2 - z_t^2)\Big)$$

$$\le \sum_{t=t_1}^{t_m-1} \frac{\eta_t}{2} \left\langle u_t - x'_{t+1}, \nabla_t - m_t \right\rangle^2 + O\Big(\sum_{t=t_1}^{t_m-1} \eta_t D_t^2 (z_{t+1}^2 - z_t^2)\Big) \qquad \text{(since } \|u_t\|_2 \le \|D_t\|_2\text{)}$$

$$\le \sum_{t=t_1}^{t_m-1} \frac{\eta_t}{2} \left\langle u_t - x'_{t+1}, \nabla_t - m_t \right\rangle^2 + O\Big(\sum_{t=t_1}^{t_m-1} D_t (z_{t+1}^2 - z_t^2)/z_t\Big) \qquad \text{(since } \eta_t \le \frac{1}{64 D_t z_t}\text{)}$$

$$\le \sum_{t=t_1}^{t_m-1} \frac{\eta_t}{2} \left\langle u_t - x'_{t+1}, \nabla_t - m_t \right\rangle^2 + O(D_{t_m} z_{t_m}^2) \qquad \text{(since } z_t \ge z_1 = 1\text{)}$$

$$\le \sum_{t=t_1}^{t_m-1} 8\eta_t \langle u_t, \widetilde{g}_t - m_t \rangle^2 + 8\eta_t \langle x_t, \widetilde{g}_t - m_t \rangle^2 + O\Big(\frac{r \ln(T\eta_{t_1} z_{t_m}/z_{t_1})}{\eta_{t_m}} + D_{t_m} z_{t_m}^2\Big).$$

**Term** $\sum_{t=t_1}^{t_m} \big(D_{\psi_t}(x_t, x'_{t+1}) + D_{\psi_t}(x_t, x'_t)\big)$:

$$\sum_{t=t_1}^{t_m} (D_{\psi_t}(x_t, x'_{t+1}) + D_{\psi_t}(x_t, x'_t)) = \sum_{t=t_1}^{t_m} \frac{1}{2}(\|x_t - x'_t\|_{A_t}^2 + \|x'_{t+1} - x_t\|_{A_t}^2)$$

$$\ge \frac{1}{2} \sum_{t=t_1+1}^{t_m} \|x_t - x'_t\|_{A_{t-1}}^2 + \frac{1}{2} \sum_{t=t_1+1}^{t_m+1} \|x_{t-1} - x'_t\|_{A_{t-1}}^2$$

$$\ge \frac{4 z_{t_1}^2}{4} \sum_{t=t_1+1}^{t_m} \|x_t - x_{t-1}\|_2^2 = z_1^2 \sum_{t=t_1+1}^{t_m} \|x_t - x_{t-1}\|_2^2,$$

where the last inequality comes from $A_t \succeq A_{t-1} \succeq 4 z_{t_1}^2 I$ when $t \in [t_1 + 1, t_m]$.

Then, we have

$$\sum_{t \in I_m} \langle \widetilde{g}_t, x_t - u_t \rangle$$

$$\le O\Big(\frac{r \ln(T\eta_{t_1} z_{t_m}/z_{t_1})}{\eta_{t_m}} + z_{t_1}^2 \|u_t\|_2^2 + D_t z_{t_m}^2 + \sum_{t=t_1}^{t_m-1} \eta_t \langle u_t, \widetilde{g}_t - m_t \rangle^2 - z_{t_1}^2 \sum_{t_1+1}^{t_m} \|x_t - x_{t-1}\|_2^2\Big)$$

$$\le \widetilde{O}\Big(\sqrt{\sum_{t \in I_m} \|\widetilde{g}_t - m_t\|_2^2} + z_{t_1}^2 \|u\|_2^2 + z_{t_m}^2 D_t + \|u\|_2^2 \sqrt{\sum_{t \in I_m} \|\widetilde{g}_t - m_t\|_2^2} - z_{t_1}^2 \sum_{t_1+1}^{t_m} \|x_t - x_{t-1}\|_2^2\Big).$$

Furthermore, by $\|\widetilde{g}_t - m_t\|_2 \leq \|g_t - m_t\|_2$, we have

$$\sum_{t \in I_m} \langle g_t - \widetilde{g}_t, x_t - u_t \rangle \leq (D_t + \|u\|_2) \sum_{t \in I_m} \|g_t - \widetilde{g}_t\| \leq (D_t + \|u\|_2) \sum_{t \in I_m} \frac{B_t - B_{t-1}}{B_t} \|g_t - m_t\|_2$$
$$\leq 2(D_t + \|u\|_2)G.$$

At the last iteration $T$, we have

$$D_T < \sqrt{\sum_{t=1}^{T} \frac{\|g_t\|_2}{\max\{1, \max_{k \leq t} \|g_k\|_2\}}}$$

$$\leq \sqrt{\sum_{t=1}^{T} \|g_t\|_2}$$

$$\leq \sqrt{\sum_{t=1}^{T} \|g_t - \mathbb{E}[g_t]\|_2 + \|\nabla F_t(x_t)\|_2}$$

$$= \sqrt{\sum_{t=1}^{T} \|\nabla f_t(x_t) - \nabla F_t(x_t)\|_2 + \|\nabla F_t(x_t)\|_2}.$$

Therefore, we have

$$\sum_{m=1}^{M} \sum_{t \in I_m} \langle g_t, x_t - u_t \rangle \leq \widetilde{O}\Big(\|u\|_2^2 \sqrt{\sum_{t \in [T]} \|g_t - m_t\|_2^2} + G^2 D_T + G^2 \|u\|_2^2 - z_1^2 \sum_{t=2}^{T} \|x_t - x_{t-1}\|_2^2\Big)$$

$$\leq \widetilde{O}\Big(\|u\|_2^2 \sqrt{\sum_{t \in [T]} \|g_t - m_t\|_2^2} + G^2 D_T + G^2 \|u\|_2^2 - \sum_{t=2}^{T} \|x_t - x_{t-1}\|_2^2\Big). \qquad \text{(since } z_1 = B_0 = 1)$$

Now, we bound $T_{\text{extra}}$. We observe that $u_t$ is either $u$ or $\frac{D_t}{\|u\|_2} u$. When $u_t \neq u$, $\|u\|_2 \geq D_t > \sqrt{\sum_{s=1}^{t} \frac{\|g_s\|_2}{\max_{k \leq s} \|g_k\|_2}}$. Once $u_t = u$, it stays there. Let $t^*$ be the last round when $u_t \neq u$.

$$\sum_{t=1}^{T} \langle g_t, u_t - u \rangle = \sum_{t=1}^{t^*} \langle g_t, u_t - u \rangle \leq \sum_{t=1}^{t^*-1} \langle g_t, u_t - u \rangle + 2\|u\|_2 G$$

$$\leq 2\|u\|_2 \sum_{t=1}^{t^*-1} \|g_t\|_2 + 2\|u\|_2 G$$

$$\leq 2\|u\|_2 G \sum_{t=1}^{t^*-1} \frac{\|g_t\|_2}{\max_{k \leq t^*-1} \|g_k\|_2} + 2\|u\|_2 G$$

$$\leq 2\|u\|_2^3 G + 2\|u\|_2 G.$$

Therefore, we have

$$\mathfrak{R}_T(u) \leq \widetilde{O}\Big(\|u\|_2^2 \sqrt{\sum_{t \in [T]} \|g_t - m_t\|_2^2} + G^2 D_T + G^2 \|u\|_2^2 + G\|u\|_2^3 - \sum_{t=2}^{T} \|x_t - x_{t-1}\|_2^2\Big)$$

By Lemma C.5, we have

$$\|u\|_2^2 \sqrt{\sum_{t=1}^{T} \|g_t - m_t\|_2^2} - \sum_{t=2}^{T} \|x_t - x_{t-1}\|_2^2$$

$$\leq G^2 \|u\|_2^2 + 2\|u\|_2^2 \sqrt{\sum_{t=2}^{T} \|\nabla F_t(x_{t-1}) - \nabla F_{t-1}(x_{t-1})\|_2^2}$$

$$+ 2\sqrt{2}\|u\|_2^2 \sqrt{\sum_{t=1}^{T} \|\nabla f_t(x_t) - \nabla F_t(x_t)\|_2^2} + 2L\|u\|_2^2 \sqrt{\sum_{t=2}^{T} \|x_t - x_{t-1}\|_2^2} - \sum_{t=2}^{T} \|x_t - x_{t-1}\|_2^2$$

$$\leq G^2 \|u\|_2^2 + L^2 \|u\|_2^4 + 2\|u\|_2^2 \sqrt{\sum_{t=2}^{T} \|\nabla F_t(x_{t-1}) - \nabla F_{t-1}(x_{t-1})\|_2^2}$$

$$+ 2\sqrt{2}\|u\|_2^2 \sqrt{\sum_{t=1}^{T} \|\nabla f_t(x_t) - \nabla F_t(x_t)\|_2^2}$$

Therefore, we can conclude that

$$\mathbb{E}[\mathfrak{R}_T(u)] \leq \widetilde{O}\Big(\|u\|_2^2(\sqrt{\sigma_{1:T}^2} + \sqrt{\Sigma_{1:T}^2}) + G^2\sqrt{\sigma_{1:T} + \mathfrak{G}_{1:T}} + G^2\|u\|_2^2 + \|u\|_2^4 + G\|u\|_2^3\Big),$$

where $\sigma_{1:T}$ captures the stochastic gradient deviation (without the squared norm) and $\mathfrak{G}_{1:T}$ denotes the sum of maximum expected gradients. $\qquad\square$

