# OpenReview forum: "Parameter-free Algorithms for the Stochastically Extended Adversarial Model"
_NeurIPS.cc/2025/Conference — NeurIPS 2025 poster_

### Official Review · Reviewer_7Ufi · 2025-07-01

**Clarity:** 3
**Significance:** 2
**Originality:** 2
**Rating:** 4
**Confidence:** 4

**Summary:**

This paper studies the stochastically extended adversarial (SEA) model in the online parameter-free setting where the domain (equivalently, the norm of the comparator) is unbounded and a uniform upper bound on the Lipschitz constants of the cost functions is not known a priori. The authors use the Optimistic Online Newton Step as the base algorithm and adapt it to the parameter-free setting by running multiple copies of base learners and combining their predictions using a multiplicative weight (MsMwC) algorithm. The authors adequately survey the related literature and put their contributions into an appropriate context.

**Questions:**

1.	Please make it clear in the problem statement (Section 2.1) whether the distribution $D_t$, from which the cost functions are sampled, is known to the learner before it selects the action $x_t$ for round t?
2.	As a follow-up to the above question, if $D_t$ is unknown and is never “exploited” by the algorithm, then how does stochasticity help in this setting?
3.	In the definition of cumulative adversarial variation (line 128), what randomness does the outer expectation refer to? The expected gradients seemed to be deterministic (c.f. Assumption 2.3).
4.	Do the results hold when the distributions $D_t$’s are chosen adaptively?
5.	No lower bound is established for the regret bounds derived in this paper. Thu,s it is not clear whether the regret bounds are tight.
6.	Theorem 4.1 “improves” upon [27, Theorem 5] by moving the expectation outside of the sup over the worst-case conditional variance. However, it does so only at the expense of an extra $||u||^2$ term. The authors discuss this limitation in Remark 4.2, however this discussion is very vague and can be classified as a conjecture. Since this “improvement” is claimed as a major contribution of the paper, I would highly recommend the authors give formal proof for the matching lower bound.

**Ethical Concerns:**

["NO or VERY MINOR ethics concerns only"]

**Final Justification:**

The authors have addressed my concerns. I will keep my score.

**Limitations:**

The paper does not have any direct negative societal impact.

**Quality:**

3

**Strengths And Weaknesses:**

The paper is well-written and easy to follow. On the flip side, as discussed in the comments below, it is not clear whether the bounds derived in this paper are tight. Please see the questions below.

---

> ### Author Rebuttal · Authors · 2025-07-30
>
> We thank the reviewer for your positive review and suggestions. Below we respond to the comments in Question (Q).
>
> ---
>
> >**Q1. Please make it clear in the problem statement (Section 2.1) whether the distribution $D_t$, from which the cost functions are sampled, is known to the learner before it selects the action for round t?**
>
> The distribution $D_t$ is NOT known to the learner before it chooses its action $x_t$. To make it concrete, we provide the following example:
>
> - The learner's action $x$ represents the parameters of a model.
> - The distribution $D_t$ is the true (and unknown) distribution over data samples, e.g., feature-label pairs $z$.
> - The **expected loss** (or true risk) is the quantity we wish to minimize: $F_t(x)=\mathbb{E}_{z\sim D_t}[\ell (x,z)]$, where $\ell (x,z)$ is the loss of the model on $z$. The learner never gets to see $F_t(x)$ directly.
> - The **realized loss function**, which is what the learner actually observes at round $t$, is based on a single sample $z_t\sim D_t$. That is, $f_t(x)=\ell(x,z_t)$.
>
> In this setting, the learner never directly accesses or knows the full distribution $D_t$. Its only information comes from observing the sequence of realized loss functions $f_1,f_2,\cdots,f_T$, one after another.
>
> >**Q2. As a follow-up to the above question, if $D_t$ is unknown and is never “exploited” by the algorithm, then how does stochasticity help in this setting?**
>
> The algorithm never knows or explicitly "exploits" the $D_t$. Let's revisit the concrete example from the response to Q1:
>
> - The learner's goal is to minimize the unknown **expected loss** $F_t(x)=\mathbb{E}_{z\sim D_t}[\ell (x,z)]$
> - The information it actually observes at round is the **realized loss** $f_t(x)=\ell(x,z_t)$
>
> In our setting, we can observe information of $f_t(x_t)$ as well as its gradient, where the stochasticity helps in this setting.
>
> >**Q3. In the definition of cumulative adversarial variation (line 128), what randomness does the outer expectation refer to? The expected gradients seemed to be deterministic (c.f. Assumption 2.3).**
>
> The outer expectation $\mathbb{E}[·]$ in the definition of cumulative adversarial variation $\Sigma_{1:T}$  refers to the **randomness in the environment's choice of the sequence of distributions** $D_1, D_2, ..., D_T$.
>
> Our framework is designed to handle the more general and challenging case of an adaptive adversary. Here, the environment's choice of the distribution $D_t$ at round $t$ can be a random variable that depends on the learner's past actions $(x_1,\cdots,x_{t-1})$. Because the choice of $D_t$ is itself a random process, the resulting expected function $F_t(x)$ and its gradient $\nabla F_t(x)$ are also random variables.
>
> We will add a **Footnote** to mention that ‘’If the environment is oblivious,  $F_t(x)$ and $\sigma_{t}^2$ become fixed (non‑random), hence the expectation operators in the definition of $\sigma_{1:T}^2$ and $\Sigma_{1:T}^2$ can be dropped.''
>
> >**Q4. Do the results hold when the distributions $D_t$’s are chosen adaptively?**
>
> As we discussed in the response to Q3, our results hold even when the distributions $D_t$ are chosen adaptively by an adversary. The adversary is permitted to choose the distribution $D_t$ at round $t$ based on the entire history of the game up to that point, including all of the learner's past actions $x_1, ..., x_{t-1}$ and the past realized loss functions $f_1, ..., f_{t-1}$.
>
> >**Q5. No lower bound is established for the regret bounds derived in this paper. Thus, it is not clear whether the regret bounds are tight.**
>
> To the best of our knowledge, a formal lower bound for the parameter-free Stochastically Extended Adversarial (SEA) model has not yet been established. Proving such a bound is a highly non-trivial task, as it requires simultaneously addressing the hardness inherent in both parameter-free learning and the SEA model itself.
>
> Let's break down the challenges:
>
> - **The Hardness of Parameter-Free Learning:** Even in the standard adversarial OCO setting, moving to a parameter-free framework introduces fundamental hardness, especially in the unbounded domain setting.
> - **The Hardness of the SEA Model:** The SEA model introduces its own layer of complexity. A lower bound for SEA (even with known parameters) must construct a sequence of functions that is difficult to learn. In [Sachs et al., (2023)], the lower bound of the stochastic part is established by showing the contradiction that any algorithm achieving a rate better than $\Omega(D\sigma\sqrt{T})$ could be converted, via an online-to-batch conversion, into an offline stochastic optimization algorithm that violates the well-known $\Omega(D\sigma\sqrt{T})$ lower bound for that setting. However, this proof relies on the assumption of a **known, bounded domain** with diameter $D$. It is not clear how this reduction-based argument could be extended to our **unconstrained (unbounded) and parameter-free setting**, where $D$ is unknown and potentially infinite.
>
>     We have indeed contemplated this challenging question in our paper, and our thoughts on a related aspect are partially reflected in **Remark 4.2 (lines 262-264)** of our paper. In that remark, we hypothesize that:
>
> > "...the $||u||^2$ term may be unavoidable in gradient-variation regret for unconstrained online learning, paralleling how the $d_0^2$ term also appears in the accelerated rate of unconstrained offline optimization."
>
> This remark connects the $||u||^2$ dependence in our online regret bound to the unavoidable $d_0^2$ dependence in offline accelerated optimization methods. This parallel suggests that the dependence on the squared norm of the comparator might be a fundamental aspect of the problem's hardness in the unconstrained setting, rather than just an artifact of a specific algorithm or analysis.
>
>
> Hence,  we would like to say constructing a single lower bound for the **parameter-free SEA** setting would require weaving these two sources of hardness together into one difficult problem instance, which is a significant undertaking. Making progress on this undertaking requires first solving each of the two problems above.
>
> >**Q6. Theorem 4.1 “improves” upon [27, Theorem 5] by moving the expectation outside of the sup over the worst-case conditional variance. However, it does so only at the expense of an extra $||u||^2$ term. The authors discuss this limitation in Remark 4.2, however this discussion is very vague and can be classified as a conjecture. Since this “improvement” is claimed as a major contribution of the paper, I would highly recommend the authors give formal proof for the matching lower bound.**
>
> From Theorem 5 in [Jacobsen & Cutkosky (2022)], we can directly obtain the expected regret bound as equation (2) scaling with $\tilde{\sigma}\_{1:T}^2$ (expectation of supremum). Throughout our paper, we are dedicated to deriving the bound in terms of $\sigma\_{1:T}^2$ (supremum of expectation). From Jensen's inequality, the expectation of the supremum is obviously larger than the supremum of the expectation. To make it concrete, we provide the following example to show that the difference between $\sigma^2\_{1:T}$ and $\tilde{\sigma}^2_{1:T}$ can be arbitrarily large.
>
> Let the decision space $X$ be the union of $n$ disjoint intervals, $X = \bigcup_{i=1}^n X_i$, where for instance $X_i = [i-1, i)$. For each round $t \in [T]$, we define the loss functions as follows:
>
> $f_t(x)=\langle c_{t,i}s_{t,i}, x\rangle, \text{if}\ x\in X_i$, where $c_{t,i} \in \set{0, 1}$ for each  $i \in [n]$ with $P(c_{t,i}=1) = 1/n$ and $P(c_{t,i}=0) = 1 - 1/n$ and $s_{t,i} \in \set{-1, 1}$ for each $i \in [n]$ with $P(s_{t,i}=1) = P(s_{t,i}=-1) = 1/2$.
>
> Then we have $\nabla f_t(x)=c_{t,i}s_{t,i}, \text{if}\ x\in X_i$ and $\nabla F_t(x)=0$.
>
> Hence,  $\sigma_t^2 = \sup_{x \in X} \mathbb{E}[||\nabla f_t(x) - \nabla F_t(x)||^2] = \sup_{x \in X} \mathbb{E}[||\nabla f_t(x)||^2]=1/n$ and $\sigma_{1:T}^2=T/n$.
>
> On the other hand, $\tilde{\sigma}\_t^2 = \mathbb{E}[\sup_{x \in X} ||\nabla f_t(x) - \nabla F_t(x)||^2] = \mathbb{E}[\sup_{x \in X} ||\nabla f_t(x)||^2]=\sup_{i \in [n]} (s_{t,i}c_{t,i})^2=\sup_{i \in [n]} c_{t,i}$,  where the last step holds because $s_{t,i}^2=1$ and $c_{t,i} \in \set{0,1}$ implies $c_{t,i}^2=c_{t,i}$.
>
> Let $Y_t = \sup_{i \in [n]} c_{t,i}$. The random variable $Y_t$ is 1 if at least one $c_{t,i}$ is 1, and 0 otherwise.
>
> $P(Y_t=0) = P(\text{all } c_{t,i} = 0) = (1 - 1/n)^n$,
>
> $P(Y_t=1) = 1 - P(Y_t=0) = 1 - (1 - 1/ n)^n.$
>
> $\tilde{\sigma}_t^2 = \mathbb{E}[Y_t] = 1 \cdot P(Y_t=1) + 0 \cdot P(Y_t=0) =  1 - \left(1 - \frac{1}{n}\right)^n.$
>
> For large $n$, we have
> $\tilde{\sigma}^2\_{1:T} = \sum_{t=1}^T \tilde{\sigma}\_t^2 \approx \sum_{t=1}^T (1-1/e) = T \cdot (1-1/e) = \Theta(T).$
>
> In this example, we can see when $n \to \infty$, $\sigma^2_{1:T}\to 0$ and $\tilde{\sigma}^2_{1:T} = \Theta(T)$.  Also, the bound presented in equation (2) becomes tighter only when $||u||_2 > \sqrt{T}$. However, in this regime, the entire bound scales as $O(T)$, which makes the regret bound even meaningless.
>
> ---
>
> [Jacobsen & Cutkosky (2022)] Andrew Jacobsen and Ashok Cutkosky. Parameter-free mirror descent. In Conference on Learning Theory, pages 4160–4211. PMLR, 2022.

---

> > ### Comment · Reviewer_7Ufi · 2025-08-01
> > **Thanks for your response**
> >
> > I would like to thank the authors for their responses. The responses did not significantly change my perspective on the paper, thus, I will keep my current evaluation.

---

> > > ### Author Response · Authors · 2025-08-01
> > > **Thanks for your reply**
> > >
> > > Thank you for providing insightful comments and valuable suggestions on our paper.

---

### Official Review · Reviewer_nf4u · 2025-07-02

**Clarity:** 4
**Significance:** 2
**Originality:** 3
**Rating:** 4
**Confidence:** 3

**Summary:**

This paper proposes a parameter-free stochastic online optimization algorithm based on optimistic online Newton step. The algorithm does not rely on problem parameters such as the Lipschitz constant and domain diameter. The regret bound of the algorithm is similar to existing algorithms that need the knoweldge of the problem parameters.

**Questions:**

1. As discussed in Weakness, The authors are suggested to give some examples, where the regret bound in [27] is much larger (in order sense) than the one in this paper.

2. Justification of the technical challenge of stochastic OCO compared to deterministic OCO

**Ethical Concerns:**

["NO or VERY MINOR ethics concerns only"]

**Final Justification:**

The authors have responded to my comments. I would like to maintain my positive rating.

**Limitations:**

Not applicable.

**Quality:**

3

**Strengths And Weaknesses:**

Strength:
1. Well written paper, which is easy to understand.
2. Clear contribution: problem-paraemter-free algorithm with regret bound similar to those requiring problem parameters

Weakness:
1. A problem-parameter-free algorithm was developed in [27]. The authors tried to justify the difference of the proposed algorithm and the one in [27]. The main difference in terms of regret bound is that the one in [27] depends on $\tilde{\sigma}^2$ (expectation of supremum), which is generally larger than $\sigma^2$ (supremum of expectation). Thought the former is obviously larger than the latter, I still doubt how much difference this will make. The authors are suggested to give some examples, where the regret bound in [27] is much larger (in order sense) than the one in this paper.

2. In general, compared to deterministic OCO, what is the main technical challenge of the considered stochastic OCO? In determinic case, the loss functions are changing, while in the stochastic case, the distributions of the loss functions are changing. Except for a variance term for each loss function, what will be the main novelty and technical challenge of stochastic OCO compared to the deterministic one?

---

> ### Author Rebuttal · Authors · 2025-07-30
>
> We thank the reviewer for your positive review and suggestions. Below we respond to the comments in Weakness (W).
>
> ---
>
> >**W1. A problem-parameter-free algorithm was developed in [27]. The authors tried to justify the difference of the proposed algorithm and the one in [27]. The main difference in terms of regret bound is that the one in [27] depends on $\tilde{\sigma}^2$(expectation of supremum), which is generally larger than $\sigma^2$ (supremum of expectation). Thought the former is obviously larger than the latter, I still doubt how much difference this will make. The authors are suggested to give some examples, where the regret bound in [27] is much larger (in order sense) than the one in this paper.**
>
> We thank the reviewer for this suggestion, which helps to further highlight the strengths of the results in this paper. We show in the following example that the difference between $\sigma^2_{1:T}$ and $\tilde{\sigma}^2_{1:T}$ can, in fact, be arbitrarily large.
>
> Let the decision space $X$ be the union of $n$ disjoint intervals, $X = \bigcup_{i=1}^n X_i$, where for instance $X_i = [i-1, i)$. For each round $t \in [T]$, we define the loss functions as follows:
>
> $f_t(x)=\langle c_{t,i}s_{t,i}, x\rangle, \text{if}\ x\in X_i$, where $c_{t,i} \in \set{0, 1}$ for each  $i \in [n]$ with $P(c_{t,i}=1) = 1/n$ and $P(c_{t,i}=0) = 1 - 1/n$ and $s_{t,i} \in \set{-1, 1}$ for each $i \in [n]$ with $P(s_{t,i}=1) = P(s_{t,i}=-1) = 1/2$.
>
> Then we have $\nabla f_t(x)=c_{t,i}s_{t,i}, \text{if}\ x\in X_i$ and $\nabla F_t(x)=0$.
>
> Hence,  $\sigma_t^2 = \sup_{x \in X} \mathbb{E}[||\nabla f_t(x) - \nabla F_t(x)||^2] = \sup_{x \in X} \mathbb{E}[||\nabla f_t(x)||^2]=1/n$ and $\sigma_{1:T}^2=T/n$.
>
> On the other hand, $\tilde{\sigma}\_t^2 = \mathbb{E}[\sup_{x \in X} ||\nabla f_t(x) - \nabla F_t(x)||^2] = \mathbb{E}[\sup_{x \in X} ||\nabla f_t(x)||^2]=\sup_{i \in [n]} (s_{t,i}c_{t,i})^2=\sup_{i \in [n]} c_{t,i}$,  where the last step holds because $s_{t,i}^2=1$ and $c_{t,i} \in \set{0,1}$ implies $c_{t,i}^2=c_{t,i}$.
>
> Let $Y_t = \sup_{i \in [n]} c_{t,i}$. The random variable $Y_t$ is 1 if at least one $c_{t,i}$ is 1, and 0 otherwise.
>
> $P(Y_t=0) = P(\text{all } c_{t,i} = 0) = (1 - 1/n)^n$,
>
> $P(Y_t=1) = 1 - P(Y_t=0) = 1 - (1 - 1/ n)^n.$
>
> $\tilde{\sigma}_t^2 = \mathbb{E}[Y_t] = 1 \cdot P(Y_t=1) + 0 \cdot P(Y_t=0) =  1 - \left(1 - \frac{1}{n}\right)^n.$
>
> For large $n$, we have
> $\tilde{\sigma}^2\_{1:T} = \sum_{t=1}^T \tilde{\sigma}\_t^2 \approx \sum_{t=1}^T (1-1/e) = T \cdot (1-1/e) = \Theta(T).$
>
> In this example, we can see when $n \to \infty$, $\sigma^2_{1:T}\to 0$ and $\tilde{\sigma}^2_{1:T} = \Theta(T)$.
>
> **In the final version, we will add a footnote at the bottom of Page 4 to mention that “The difference between $\sigma^2_{1:T}$ and $\tilde{\sigma}^2_{1:T}$ can, in fact, be arbitrarily large.” Also, we will add the details of the above example as a new Appendix subsection.**
>
> >**W2. In general, compared to deterministic OCO, what is the main technical challenge of the considered stochastic OCO? In determinic case, the loss functions are changing, while in the stochastic case, the distributions of the loss functions are changing. Except for a variance term for each loss function, what will be the main novelty and technical challenge of stochastic OCO compared to the deterministic one?**
>
> Thanks for your comment. From a certain perspective, the stochastic online convex optimization (SCO) problem setting might appear easier than the adversarial (or deterministic) OCO problem. In the deterministic case, the sequence of loss functions can be chosen by an adversary to be worst-case at every step. In contrast, SCO assumes the loss functions are drawn i.i.d. from a fixed, underlying distribution, which provides a stable structure to the problem. According to [Sachs, et al. 2022], the SEA model provides a unifying framework that encompasses a range of important learning models. This not only includes the fully adversarial model and the stochastic i.i.d. model, but also extends to include the adversarially corrupted i.i.d. model and Random Order Models (ROM) as special cases.
>
> Nonetheless, there is a rich and distinct body of research for SCO. The core technical challenge and novelty arise because **simply applying methods designed for the worst-case deterministic OCO setting to solve SCO problems can be suboptimal.** A generic deterministic algorithm is built to be robust against any possible sequence of losses and is therefore unable to exploit the statistical regularity inherent in the i.i.d. assumption of SCO. The innovation in SCO lies in designing algorithms that leverage this statistical structure to achieve much faster convergence rates for minimizing the *expected risk.*
>
> This suboptimality is most evident under stronger assumptions like strong convexity:
>
> - **For strongly convex and non-smooth functions**, a standard online-to-batch conversion of a deterministic algorithm with $O(\log T)$  regret would yield an $O(\log T/T)$ rate. However, specialized SCO algorithms can achieve a faster $O(1/T)$ rate.
> - **For strongly convex and smooth functions:** Deterministic OCO methods cannot break the fundamental regret barriers imposed by the adversarial setting. In contrast, by exploiting both smoothness and strong convexity in the stochastic setting, it's possible to achieve **convergence rates** $e^{-T/\kappa}$. A simple online-to-batch conversion of a deterministic algorithm cannot produce such an exponential convergence rate.
>
> Our paper's primary focus is on the general convex case. In this setting, the rates for SCO and deterministic OCO are both typically $O(1/\sqrt{T})$, making the performance gap less pronounced than in the strongly convex case.  Our results for the SEA model, which bridges these two worlds, demonstrate how to achieve near-optimal rates that adapt to the problem's structure—from the purely stochastic to the fully adversarial—without needing to know this structure in advance. Crucially, we show that our parameter-free algorithm can be obtained without loss in performance for the general convex case.

---

> > ### Comment · Reviewer_nf4u · 2025-08-05
> >
> > Thank you for addressing my comments. I will keep my positive rating.

---

### Official Review · Reviewer_hGBe · 2025-07-02

**Clarity:** 3
**Significance:** 3
**Originality:** 3
**Rating:** 5
**Confidence:** 4

**Summary:**

The paper considers Stochastically Extended Adversarial (SEA) model, a framework that bridges adversarial and stochastic online convex optimization, and focuses on designing parameter-free algorithms, which refers to algorithms that requires no prior information about the domain $D$ and the Lipschitz constant $G$. The authors first develop a Optimistic Online Newton Step (OONS) algorithm and establish a comparator-adaptive regret bound, and then generalize their algorithm to the parameter-free setting where $D$ and $G$ are unknown.

**Questions:**

1. See weakness.
2. Is the main difference between the proposed OONS and the existing OOMD [1] in this paper just in the regularization, or are there other important differences between these two algorithms?
3. Is there any lower bound for parameter-free SEA?



[1] Chen, S., Zhang, Y. J., Tu, W. W., Zhao, P., & Zhang, L. (2024). Optimistic online mirror descent for bridging stochastic and adversarial online convex optimization. Journal of Machine Learning Research, 25(178), 1-62.

**Ethical Concerns:**

["NO or VERY MINOR ethics concerns only"]

**Final Justification:**

Most of my concerns are resolved.

**Limitations:**

yes

**Paper Formatting Concerns:**

NO formatting concerns

**Quality:**

3

**Strengths And Weaknesses:**

**Strengths**

The paper is well written. The OONS algorithm is novel. The authors clearly demonstrate the advantages of OONS compared to existing optimism online mirror descent algorithms. The proof is clear and easy to follow.

**Weaknesses**

1. The biggest issue is that the parameter-free regret bound is not worst-case optimal. When D is unknown but G is known, the algorithm’s regret is larger by a factor of $||u||_2$ compared to the setting where both D and G are known. This makes the results of this paper insufficiently strong, especially considering that in the parameter-free online learning setting, existing algorithms [3] can achieve adaptive regret with worst-case optimality.
2. Although the authors have eliminated the dependence on D and G, their algorithm is still not fully parameter-free, as it requires knowing the value of $T$ in advance. In previous parameter-free online learning algorithms [1][2], it was typically not necessary to know $T$ in advance.
3. ONS is a second-order method, which can be computationally inefficient in practice.





[1] Cutkosky, A. (2019, June). Artificial constraints and hints for unbounded online learning. In Conference on Learning Theory (pp. 874-894). PMLR.

[2] Cutkosky, A., & Orabona, F. (2018, July). Black-box reductions for parameter-free online learning in banach spaces. In Conference On Learning Theory (pp. 1493-1529). PMLR.

[3] Jacobsen, A., & Cutkosky, A. (2022, June). Parameter-free mirror descent. In Conference on Learning Theory (pp. 4160-4211). PMLR.

---

> ### Author Rebuttal · Authors · 2025-07-30
>
> We thank the reviewer for your positive review and suggestions. Below we respond to the comments in Weakness (W) and Questions (Q).
>
> ---
>
> >**W1. The parameter-free regret bound is not worst-case optimal.**
>
> The main contribution of our work is to develop the first parameter-free algorithms for the SEA model. The key challenge here is not just to be free of parameters $D$ and $G$, but to achieve a regret bound that adapts to the problem's underlying statistical structure, as captured by the cumulative stochastic variance $\sigma^2_{1:T}$  and cumulative adversarial variation $\Sigma^2_{1:T}$. As we mention in Section 2.2, from Theorem 5 in [Jacobsen & Cutkosky (2022)] we can directly obtain the expected regret bound as equation (2) (but scaling with $\tilde{\sigma}\_{1:T}^2$), where it does not contain the additional $||u||\_2^2$. But $\tilde{\sigma}\_{1:T}^2$ can be arbitrarily larger than $\sigma^2\_{1:T}$ in some cases.  In the **Key Challenge** part, we discuss the technical difficulty that lies in how to achieve this preferred $\sigma\_{1:T}^2$ scaling without knowledge of $D$ and $G$ for unbounded domains. Also, in Remark 4.2, we discuss that the dependency on $||u||_2^2$ may be unavoidable.
>
> >**W2. The algorithm is still not fully parameter-free, as it requires knowing the value of $T$ in advance. In previous parameter-free online learning algorithms [1][2], it was typically not necessary to know in advance.**
>
> In the context of this paper and much of the related literature on Comparator- and Lipschitz-adaptive learning, the term "parameter-free" is primarily used to signify independence from the two most critical and often unknown parameters:
>
> - **The domain diameter** $D$**:** Can be infinite in unconstrained settings.
> - **The Lipschitz constant (or gradient bound)** $G$
>
> When we named the algorithm, we also used Comparator- and Lipschitz-adaptive. Our work successfully eliminates the need for prior knowledge of $D$ and $G$.
>
> As presented in Section 4.1, our algorithm requires prior knowledge of the time horizon $T$ to set the number of experts, $N=\lceil \log T\rceil$.  This limitation is not fundamental to our approach and can be addressed by applying a well-established technique **doubling trick**. This method is specifically designed to convert a fixed-horizon online algorithm into an **anytime** algorithm that does not need to know the total number of rounds in advance.
>
> The procedure is as follows:
>
> 1. The learning process is divided into epochs $k=1,2,3,\dotsc$. Epoch $k$  runs for $2^{k-1}$ time steps (i.e., from $t=2^{k-1}$ to $t=2^k-1$ ).
> 2. At the beginning of each epoch $k$, we restart our entire algorithm (including all base learners and the meta-algorithm).
> 3. Within epoch $k$, we run our algorithm by treating the length of that epoch, $T_k=2^{k-1}$, as the "known" time horizon. This means the number of experts is set to $N_k=\lceil \log T_k\rceil$.
>
> This approach will introduce an additional logarithmic factor $\log T$ into the final regret bound. Also, the algorithm needs **restart** in this approach.
>
> Furthermore, we can adopt a more sophisticated technique similar to the one used by [**Mhammedi et al. (2019) in Section 3.2]** of their work on Lipschitz-adaptive MetaGrad.  Specifically, instead of pre-defining a fixed number of experts based on $T$, they define an **infinite, discrete grid of potential learning rates** (our "experts"). At any given time, only a finite and adaptively chosen subset of these experts is "awake" and active. An expert "wakes up" only when the cumulative statistics of the problem (e.g., the observed gradient norms) suggest that its corresponding learning rate has become relevant. This is often referred to as a sleeping expert technique.
>
> In summary, thank you for raising this point regarding the dependency on $T$. In line with the majority of the literature in this area, the primary theoretical focus of our work has been on the core challenges of achieving **comparator- and Lipschitz-adaptive learning** in the SEA model.  We will add a **Remark in Section 4.1** in the final version to explicitly state that our algorithm can be made "anytime" (i.e., independent of a known $T$) by applying a standard doubling trick or a more sophisticated sleeping expert approach. We will provide the details of the standard doubling trick as mentioned above in a new appendix.
>
> > **W3. ONS is a second-order method, which can be computationally inefficient in practice.**
>
> Thank you for raising this point about computational efficiency.
>
> - **Clarification: ONS as a Pseudo Second-Order Method:** We clarify that while ONS leverages second-order *information* by adaptively building an inverse of a local Hessian approximation, it is technically a **first-order (Pseudo Second-Order) method** in terms of its oracle requirements. It only requires access to **gradients**, not the actual Hessian matrix. The main computational task is updating and using the matrix $A_t$, not querying a second-order oracle. To be precise, the matrix $A_t$ is constructed iteratively from the outer products of past gradients, and does not require direct computation or approximation of the Hessian matrix at any step.
>
> - **Computational Cost of ONS:** The inverse calculation of the matrix $A_t$ is indeed the primary computational step. However, because $A_t = cI + \sum \eta_s v_s v_s^T$ is updated via a simple rank-1 update at each step, its inverse can be computed very efficiently using the **Sherman-Morrison-Woodbury formula**. This reduces the complexity from $O(d^3)$ to $O(d^2)$ per step, which is the same as OMD.
>
> - **Justification for Using ONS instead OMD:** Our choice of ONS was a necessary one to tackle **unconstrained domain** in the parameter-free setting. As we explicitly state in lines 201-205, the regret bound for OMD typically scales with $D$, the domain diameter. For instance, the bound often contains a term like $D\sqrt{\sum||g_t - m_t||^2}$. When $D$ is unknown and potentially infinite, this dependence becomes problematic and can lead to vacuous $O(T)$ regret bounds. The adaptive second-order information in ONS is precisely the mechanism that allows us to decouple the regret from the unknown domain size. In our analysis (see Theorem 3.1 and its proof), the dependence on $D$ is isolated to terms like $D(z_T - z_1)$.
>
> >**Q1. Is the main difference between the proposed OONS and the existing OOMD [1] in this paper just in the regularization, or are there other important differences between these two algorithms?**
>
> The main difference lies in the regularizer used at each step. In OOMD, the regularizer like the squared Euclidean norm. Our work employs a time-varying regularizer. Specifically, it uses $\frac{1}{2}||x||_{A_t}^2$, where the matrix $A_t$ is adaptively built from the observed gradients. This choice of regularizer is not arbitrary; it is the key enabler for achieving our parameter-free results, particularly comparator-adaptivity in unconstrained domains as we mentioned in the response to **W3**.
>
> >**Q2. Is there any lower bound for parameter-free SEA?**
>
> To the best of our knowledge, a formal lower bound for the parameter-free Stochastically Extended Adversarial (SEA) model has not yet been established. Proving such a bound is a highly non-trivial task, as it requires simultaneously addressing the hardness inherent in both parameter-free learning and the SEA model itself.
>
> Let's break down the challenges:
>
> - **The Hardness of Parameter-Free Learning:** Even in the standard adversarial OCO setting, moving to a parameter-free framework introduces fundamental hardness, especially in the unbounded domain setting.
> - **The Hardness of the SEA Model:** The SEA model introduces its own layer of complexity. A lower bound for SEA (even with known parameters) must construct a sequence of functions that is difficult to learn. In [Sachs et al., (2023)], the lower bound of the stochastic part is established by showing the contradiction that any algorithm achieving a rate better than $\Omega(D\sigma\sqrt{T})$ could be converted, via an online-to-batch conversion, into an offline stochastic optimization algorithm that violates the well-known $\Omega(D\sigma\sqrt{T})$ lower bound for that setting. However, this proof relies on the assumption of a **known, bounded domain** with diameter $D$. It is not clear how this reduction-based argument could be extended to our **unconstrained (unbounded) and parameter-free setting**, where $D$ is unknown and potentially infinite.
>
> We have indeed contemplated this challenging question in our paper, and our thoughts on a related aspect are partially reflected in **Remark 4.2 (lines 262-264)** of our paper. In that remark, we hypothesize that:
>
> > "...the $||u||^2$ term may be unavoidable in gradient-variation regret for unconstrained online learning, paralleling how the $d_0^2$ term also appears in the accelerated rate of unconstrained offline optimization."
>
> This remark connects the $||u||^2$ dependence in our online regret bound to the unavoidable $d_0^2$ dependence in offline accelerated optimization methods. This parallel suggests that the dependence on the squared norm of the comparator might be a fundamental aspect of the problem's hardness in the unconstrained setting, rather than just an artifact of a specific algorithm or analysis.
>
> Hence,  we would like to say constructing a single lower bound for the **parameter-free SEA** setting would require weaving these two sources of hardness together into one difficult problem instance, which is a significant undertaking. Making progress on this undertaking requires first solving each of the two problems above.
>
> ---
>
> [Mhammedi et al. (2019)] Lipschitz adaptivity with multiple learning rates in online learning. COLT, 2019.
>
> [Jacobsen & Cutkosky (2022)] Parameter-free mirror descent. COLT, 2022.

---

> ### Comment · Reviewer_hGBe · 2025-08-05
>
> Thanks for the response! Yep the double trick works in theory, but it’s not practical in real-world scenarios (not efficient). What I hope to see is a more elegant adaptive method like the one in [2].
>
> Most of my concerns are addressed. I would like to increase my score to clear accept.

---

> > ### Author Response · Authors · 2025-08-05
> > **Thanks for your follow-up**
> >
> > Thank you for this insightful follow-up and for suggesting this excellent direction. While we appreciate these adaptive methods, directly applying them to our SEA model setting while preserving the problem-dependent regret guarantees is highly non-trivial.
> >
> > The doubling trick, while theoretically valid, often comes with practical drawbacks like restarts. An elegant solution, as you suggest, would require a more sophisticated **adaptive method.** Examples of such adaptive approaches include the **coin-betting framework** and the **Lipschitz-adaptive MetaGrad** from [Mhammedi et al. (2019)]. However, these methods are significantly more non-trivial. Their analytical frameworks are tailored to different structural assumptions and are not immediately compatible with the specific demands of our SEA setting. Integrating these sophisticated approaches with our framework is a substantial undertaking that requires significant further research.
> >
> > In summary, thank you for raising these important points regarding the dependency on $T$. This is an interesting and challenging direction for future research. In our original manuscript, we also mention this point in the **Conclusion**. We also want to highlight that the primary theoretical focus of our work has been on the core challenges of achieving **comparator- and Lipschitz-adaptive learning** in the SEA model, as the name of our proposed algorithm.
> >
> > ---
> >
> > [Mhammedi et al. (2019)] Lipschitz adaptivity with multiple learning rates in online learning. COLT, 2019.

---

### Official Review · Reviewer_ygmp · 2025-07-05

**Clarity:** 3
**Significance:** 2
**Originality:** 2
**Rating:** 5
**Confidence:** 1

**Summary:**

This paper proposes the first parameter-free algorithm based on the optimistic online Newton step algorithm for the stochastically extended adversarial (SEA) model, which is a hybrid model of stochastic and adversarial frameworks for online convex optimization. The authors obtain the regret bound of the proposed algorithm for the scenario with unknown domain D but known Lipschitz constant of the loss function G, and the scenario with both G and D unknown.

**Questions:**

1. I understand this paper is a theoretical paper and acknowledge its theoretical contributions, but I wonder, is it possible to validate the efficiency of the algorithm by numerical experiments?
2. Could the authors discuss the computational cost under the high-dimensional settings?
3. I wonder what the order of the variance of the algorithm is or how the variance of the proposed algorithm compares with other non-parameter-free algorithms.

**Ethical Concerns:**

["NO or VERY MINOR ethics concerns only"]

**Final Justification:**

The paper makes solid contributions for the stochastically extended adversarial model. The authors have improved their paper by adding some discussions about the computational cost under the high-dimensional settings. Their justifications about not making numerical experiments and variance analysis are acceptable.

However, since I am not very familiar with this area. I have set the confidence as 1.

**Limitations:**

Yes

**Paper Formatting Concerns:**

Equations in the Appendix are not formatted well. For example,  the equations on Page 16 run off the page limit.

**Quality:**

3

**Strengths And Weaknesses:**

Strengths
1. The contributions are clear.
2. The technical results are solid.
3. The contributions are novel since there is no parameter-free algorithm for SEA. This paper fills this gap.

Weaknesses
The dependence of the regret-bound on the norm of the comparator is not optimal. Since the authors have discussed this point in detail, I don't see this as a big concern.

---

> ### Author Rebuttal · Authors · 2025-07-30
>
> We thank the reviewer for your positive review and suggestions. Below we respond to the comments in Questions (Q).
>
> ---
>
> >**Q1. I understand this paper is a theoretical paper and acknowledge its theoretical contributions, but I wonder, is it possible to validate the efficiency of the algorithm by numerical experiments?**
>
> Thanks for the comment. Since our contributions are primarily theoretical, we did not prioritize numerical experiments. This approach aligns with prior theoretical works in online convex optimization ( e.g., [Cutkosky (2019)], [Jacobsen & Cutkosky (2022)],  [Jacobsen & Cutkosky (2023)], [Sachs et al. (2023)] and [Chen et al. (2024)] ).
>
> >**Q2. Could the authors discuss the computational cost under the high-dimensional settings?**
>
> The primary computational bottleneck in our proposed OONS algorithm (Algorithm 1) is the management of the $d \times d$ matrix $A_t$. A naive implementation would involve storing this dense matrix and performing matrix inversion at each step, leading to prohibitive costs.
>
> - **Storage:** Storing the dense $d \times d$ matrix $A_t$ requires $O(d^2)$ memory.
> - **Computation:** A naive matrix inversion $A_t^{-1}$ would cost $O(d^3)$ per step. The matrix-vector product would cost $O(d^2)$ per step.
>
> In practice, the matrix $A_t$ is constructed by a sum of outer products, $A_t = cI + \sum \eta_s v_s v_s^T$. The matrix $A_t$ is constructed iteratively from the outer products of past gradients, and does not require direct computation or approximation of the Hessian matrix at any step. Its inverse can be efficiently computed and updated at each step using the Sherman-Morrison-Woodbury formula, reducing the complexity from $O(d^3)$ to $O(d^2)$.
>
> However, the high-dimensional settings should be tackled separately, like even for the stochastic/online optimization problem, high dimension is also an independent problem to work with. To address this, existing research has explored like:
>
> - **Matrix Sketching:** This technique approximates the original $d\times d$  matrix $A_t$  with a much smaller "sketched" matrix, thereby significantly reducing storage and computational requirements. For instance, related work [Luo et al, (2016)] has achieved matrix-free updates by applying sketching to the Online Newton Step (ONS), avoiding direct manipulation of the high-dimensional matrix.
> - **Sparsity:** If the gradient vectors are sparse in most iterations, sparse data structures and algorithms can be utilized to perform the updates, thus avoiding the full $O(d^2)$ matrix-vector multiplications.
>
> Integrating these high-dimensional adaptation techniques into our proposed algorithm for the SEA model and analyzing their theoretical guarantees is a very interesting research direction. **We will include the above complexity analysis as a subsection in the appendix of the final version. For the high-dimensional settings, we will mention this in the discussion and conclusion sections of our paper as a prospect for future work**.
>
> >**Q3. I wonder what the order of the variance of the algorithm is or how the variance of the proposed algorithm compares with other non-parameter-free algorithms.**
>
> Thank you for this insightful question. If we understand your comment correctly, you are asking about the variance of the regret, and how it compares with non-parameter-free algorithms.
>
> To the best of our knowledge, the analysis of regret variance for the SEA model is a largely unexplored area. The existing literature on this model, including the foundational non-parameter-free works in [Sachs et al. (2023)] and [Chen et al. (2024)], has primarily focused on deriving bounds for the **expected regret,** $\mathbb{E}[R_T(u)]$. These seminal works established the key dependencies on the adversarial variation and stochastic variance in expectation. Therefore, a direct comparison of our algorithm's regret variance to that of existing non-parameter-free SEA algorithms is difficult.
>
> You raise a very important point. Most current regret analyses provide guarantees on the expected regret, which is a deterministic quantity. The question of regret **variance** is deeply connected to the study of **high-probability regret bounds**. Deriving such high-probability bounds for the SEA model, particularly in the parameter-free setting, is a significant and non-trivial challenge. We believe this is an interesting direction for future research and will add a discussion of this point to our paper.
>
> ---
>
> [Cutkosky (2019)] Ashok Cutkosky. Artificial constraints and hints for unbounded online learning. In Conference on Learning Theory, pages 874–894. PMLR, 2019.
>
> [Jacobsen & Cutkosky (2022)] Andrew Jacobsen and Ashok Cutkosky. Parameter-free mirror descent. In Conference on Learning Theory, pages 4160–4211. PMLR, 2022.
>
> [Jacobsen & Cutkosky (2023)]  Andrew Jacobsen and Ashok Cutkosky. Unconstrained online learning with unbounded losses. International Conference on Machine Learning. PMLR, 2023.
>
> [Sachs et al. (2023)] Sarah Sachs, et al. Accelerated rates between stochastic and adversarial online convex optimization. arXiv preprint arXiv:2303.03272 (2023).
>
> [Chen et al. (2024)] Sijia Chen, Yu-Jie Zhang, Wei-Wei Tu, Peng Zhao, and Lijun Zhang. Optimistic online mirror descent for bridging stochastic and adversarial online convex optimization. Journal of Machine Learning Research, 2024.
>
> [Luo et al. (2016)] Haipeng Luo, et al. Efficient second-order online learning by sketching. Advances in Neural Information Processing Systems 29, 2016.

---

> > ### Comment · Reviewer_ygmp · 2025-08-05
> >
> > The authors have addressed my concern. I will keep my score.

---

### Note · Authors · 2025-08-12

We sincerely thank the Area Chair and all reviewers for the time and effort devoted to evaluating our work. We greatly appreciate the positive reviews and insightful suggestions from all reviewers.

Our main contribution is designing parameter-free algorithms that do not require prior knowledge of the domain or the Lipschitz constant (comparator- and Lipschitz-adaptive learning) for the Stochastically Extended Adversarial (SEA) model. We believe that we have addressed most of the reviewers’ concerns during the rebuttal phase.

Thank you for your time and all thoughtful comments during this process.


Sincerely,

The Authors

---

### Decision · Program_Chairs · 2025-09-17

**Decision:**

Accept (poster)

**Comment:**

This work studies the challenge of adaptivity in the Stochastically Extended Adversarial (SEA) model, a model proposed to bridge between i.i.d. and fully adversarial settings in online learning. This work is the first to provide adaptive rates in this model, which I believe makes this model much more relevant and usable in practice. Reviewers noticed some limitations, but they are overall satisfied and enthusiastic about the merits of this paper. Hence, I strongly recommend acceptance.